# A self-sustaining endocytic-based loop promotes breast cancer plasticity leading to aggressiveness and pro-metastatic behavior

Irene Schiano Lomoriello[1,2,9], Giovanni Giangreco [1,2,9], Claudia Iavarone[1,2,9], Chiara Tordonato[1,2,3], Giusi Caldieri[1,2,3], Gaetana Serio[2], Stefano Confalonieri[1,2], Stefano Freddi[1], Fabrizio Bianchi [1,5], Stefania Pirroni[1,2,6], Giovanni Bertalot[1], Giuseppe Viale[1,3], Davide Disalvatore [1,2], Daniela Tosoni[1], Maria Grazia Malabarba[1,2,3], Andrea Disanza [2], Giorgio Scita [2,3], Salvatore Pece[1,3], Brian K. Pilcher[4,7], Manuela Vecchi [1,2,8,10], Sara Sigismund [1,2,3,10✉] & Pier Paolo Di Fiore [1,2,3,10✉]

The subversion of endocytic routes leads to malignant transformation and has been implicated in human cancers. However, there is scarce evidence for genetic alterations of endocytic proteins as causative in high incidence human cancers. Here, we report that Epsin 3 (EPN3) is an oncogene with prognostic and therapeutic relevance in breast cancer. Mechanistically, EPN3 drives breast tumorigenesis by increasing E-cadherin endocytosis, followed by the activation of a β-catenin/TCF4-dependent partial epithelial-to-mesenchymal transition (EMT), followed by the establishment of a TGFβ-dependent autocrine loop that sustains EMT. EPN3-induced partial EMT is instrumental for the transition from in situ to invasive breast carcinoma, and, accordingly, high EPN3 levels are detected at the invasive front of human breast cancers and independently predict metastatic rather than loco-regional recurrence. Thus, we uncover an endocytic-based mechanism able to generate TGFβ-dependent regulatory loops conferring cellular plasticity and invasive behavior.

[1] IEO, Istituto Europeo di Oncologia IRCCS, Via Ripamonti 435, 20141 Milan, Italy. [2] IFOM, Fondazione Istituto FIRC di Oncologia Molecolare, Via Adamello 16, 20139 Milan, Italy. [3] Università degli Studi di Milano, Dipartimento di Oncologia ed Emato-oncologia, Via Santa Sofia 9/1, 20122 Milan, Italy. [4] UTSW Medical Center, Department of Cell Biology, Dallas, TX, USA. [5] Present address: Institute for Stem-cell Biology, Regenerative Medicine and Innovative Therapies (ISBReMIT), Casa Sollievo della Sofferenza Hospital – IRCCS, San Giovanni Rotondo, (FG), Italy. [6] Present address: Divisione di Anatomia e Istologia Patologica, Presidio San Giovanni Di Dio, Via Ospedale 46, Cagliari, Italy. [7] Present address: Critical Mass Scientific Strategy Consultants, Raleigh, NC, USA. [8] Present address: Cogentech S.R.L. Benefit Corporation with a Sole Shareholder, Via Adamello 16, 20139 Milan, Italy. [9] These authors contributed equally: Irene Schiano Lomoriello, Giovanni Giangreco, Claudia Iavarone. [10] These authors jointly supervised this work: Manuela Vecchi, Sara Sigismund, Pier Paolo Di Fiore. ✉email: sara.sigismund@ieo.it; pierpaolo.difiore@ieo.it

Endocytosis is a critical regulator of several cellular processes, and its subversion is thought to play a role in tumorigenesis[1,2]. Indeed, endocytic proteins, including endocytic adaptors, Rab family members, and E3 ligases are altered in different tumor types[1]. The link between endocytosis and cancer is further strengthened by the observation that altered vesicular trafficking of growth factor receptors and of integrin- and cadherin-based adhesion complexes can lead to epithelial-to-mesenchymal transition (EMT)[3]. EMT is a transcriptional program associated with the acquisition of a motile/invasive state and the emergence of cancer stem cells (CSCs), which drive tumor initiation, progression, and metastasis[4,5]. Increasing evidence supports the existence of intermediate EMT states (so-called partial EMT) characterized by the concomitant presence of both epithelial and mesenchymal features, offering a more dynamic interpretation of the fluidity and plasticity of the EMT program[6]. Indeed, intermediate EMT states have been identified as key drivers of organ development and fibrosis, as well as cancer progression[5,6].

Epsin3 (EPN3), together with Epsin1 (EPN1) and Epsin2 (EPN2), belongs to the evolutionarily conserved epsin family[7,8]. Whereas EPN1 and EPN2 are well-characterized endocytic adaptors[7], the function of EPN3 is largely unknown. *Epn3* knockout mice did not display obvious phenotypes, likely due to redundancy with other epsins[9]. Furthermore, while EPN1 and EPN2 are ubiquitous, EPN3 is expressed at low levels in normal tissues, except for gastric parietal cells[9], arguing for specialized functions. Notably, EPN3 expression is upregulated in wounded epithelial tissues (e.g., ulcerative colitis) exhibiting altered cell–extracellular matrix interactions[8]. High EPN3 levels were also detected in migrating keratinocytes in cutaneous wounds, but not in differentiating keratinocytes[8]. Finally, enforced EPN3 expression has been linked to increased cell migration[10,11].

Here, we show that EPN3 is overexpressed in ~40% of breast cancers (BCs) and that its overexpression (associated with gene amplification in ~25% of the overexpressing cases) is an independent predictor of distant metastasis. We further demonstrate that EPN3 overexpression induces a state of partial EMT (assessed by a variety of biological and biochemical phenotypes), triggered by EPN3-dependent endocytosis of ECAD and sustained through a feed-forward loop between ECAD internalization and enhanced TGFβ signaling. Finally, EPN3 protein levels are upregulated at the invasive front of human BCs that are undergoing the in situ-to-invasive transition, and its expression is required for the transition from in situ to invasive carcinomas in model systems. These results identify EPN3 as an oncogene that is frequently altered in BC and which acts as an independent predictor of disease outcome.

## Results

**The *EPN3* gene is amplified and overexpressed in BC.** *EPN3* is located on chr. 17q21.33, 10.8 Mbps from *ERBB2* (Fig. 1a). In public databases, *EPN3* is putatively amplified in ~7–8% of BCs, and co-amplified with *ERBB2* in around half of these cases (Fig. 1b, left and middle). To obtain direct evidence of *EPN3* amplification, we performed fluorescence in situ hybridization (FISH) on an independent cohort of BC patients[12,13] and found *EPN3* amplified in ~10% of the cases. *EPN3* and *ERBB2* were co-amplified in ~5% of all cases, and individually amplified in ~5% and ~13% of cases, respectively (Fig. 1b, right). By immunohistochemistry (IHC) analysis, in the same cohort of patients (Fig. 1c; Supplementary Fig. 1A), there was correspondence between *EPN3* amplification and overexpression in almost all cases (Fig. 1d). Furthermore, in more than one-third of cases, EPN3 was overexpressed in the absence of amplification (Fig. 1d).

Thus, *EPN3* is amplified and/or overexpressed in BC, and its amplification can occur independently of, or concomitantly with, *ERBB2* amplification.

**EPN3 overexpression induces partial EMT in MCF10A cells.** Initial experiments in BC cell lines (Supplementary Fig. 2A, B) revealed that silencing EPN3 reduced tumorigenicity in cells harboring *EPN3* amplification (i.e., BT474, Supplementary Fig. 2C, D). Conversely, *EPN3* ectopic overexpression in not-amplified/overexpressing BC cells (i.e., HCC1569) increased their in vivo tumorigenic potential (Supplementary Fig. 2E), arguing that EPN3 overexpression might be an advantage-conferring event in BC. Thus, we investigated the mechanisms through which EPN3 overexpression contributes to tumorigenesis.

The nontumorigenic mammary epithelial cell line, MCF10A, displays low EPN3 levels and no alterations of its locus (Supplementary Fig. 2A, B)[14]. In these cells, we overexpressed EPN3 at levels comparable with those present in the EPN3-amplified BT474 cell line (Fig. 2a). Overexpression of EPN3, but not of the related EPN1 (used as a control), induced a morphology reminiscent of EMT (Fig. 2b) and resembling that induced by the known EMT inducer, TWIST (Fig. 2b; Supplementary Fig. 3A)[15]. When EMT markers were analyzed by immunoblot (IB), MCF10A-EPN3 showed increased levels of NCAD (N-cadherin) and VIM (Vimentin), and a modest decrease of ECAD (E-cadherin) vs. MCF10A-control (Ctr) or MCF10A-EPN1 (Fig. 2c). While the increase in NCAD and VIM was paralleled by increased transcription, the decrease in ECAD did not appear to involve a transcriptional mechanism, at variance with MCF10A-TWIST (Fig. 2d). EPN3 also induced increased mRNA levels of fibronectin (*FN1*) and of the EMT master regulators, *ZEB1*, *SNAIL* and *TWIST* (Supplementary Fig. 3B). Finally, MCF10A-EPN3 displayed increased invasiveness in vitro (Fig. 2e). Thus, the EPN3-induced EMT program resembles the so-called partial EMT state, a metastable/plastic state characterized by the concomitant expression of both epithelial and mesenchymal markers[6].

When grown in 3D Matrigel, MCF10A-Ctr formed polarized, *acini*-like spheroids recapitulating aspects of glandular architecture in vivo, including the formation of a hollow lumen, apico-basal polarization, and the deposition of basement membrane components[16]. MCF10A-EPN3, similarly to MCF10A-TWIST, displayed aberrant morphogenesis, with around 40% of organoids failing to complete the differentiation program (Supplementary Fig. 3C), while the remaining ones were aberrant (Fig. 2f; Supplementary Fig. 3D) with multilobular, filled acini of irregular shape, displaying defects in cell polarization and in the polarized deposition of basement membrane (Fig. 2g).

**EPN3 induces an invasive phenotype in primary organoids.** To investigate the 3D morphogenesis alterations induced by EPN3 in a more physiological setting, we derived organoids from the mammary epithelium of conditional EPN3 knock-in (*EPN3-KI*) mice, in which the expression of human EPN3 under a strong promoter can be induced by treatment of cells with CRE recombinase (see "Methods"). These organoids, in absence of CRE treatment, closely resembled the in vivo mammary bilayered acini structures, as they contained both luminal (keratin 8, K8-positive) and myoepithelial (keratin 5, K5-positive, Supplementary Fig. 3E) cells. Upon treatment of the organoids with CRE ex vivo, they lost their confined organization and became invasive (Fig. 3a), with many cells delaminating from the organoid core and invading the surrounding Matrigel (Fig. 3a, b). Of note, these EPN3-positive invading cells displayed significant loss of their epithelial identity (i.e. they were negative for K5 and K8 and

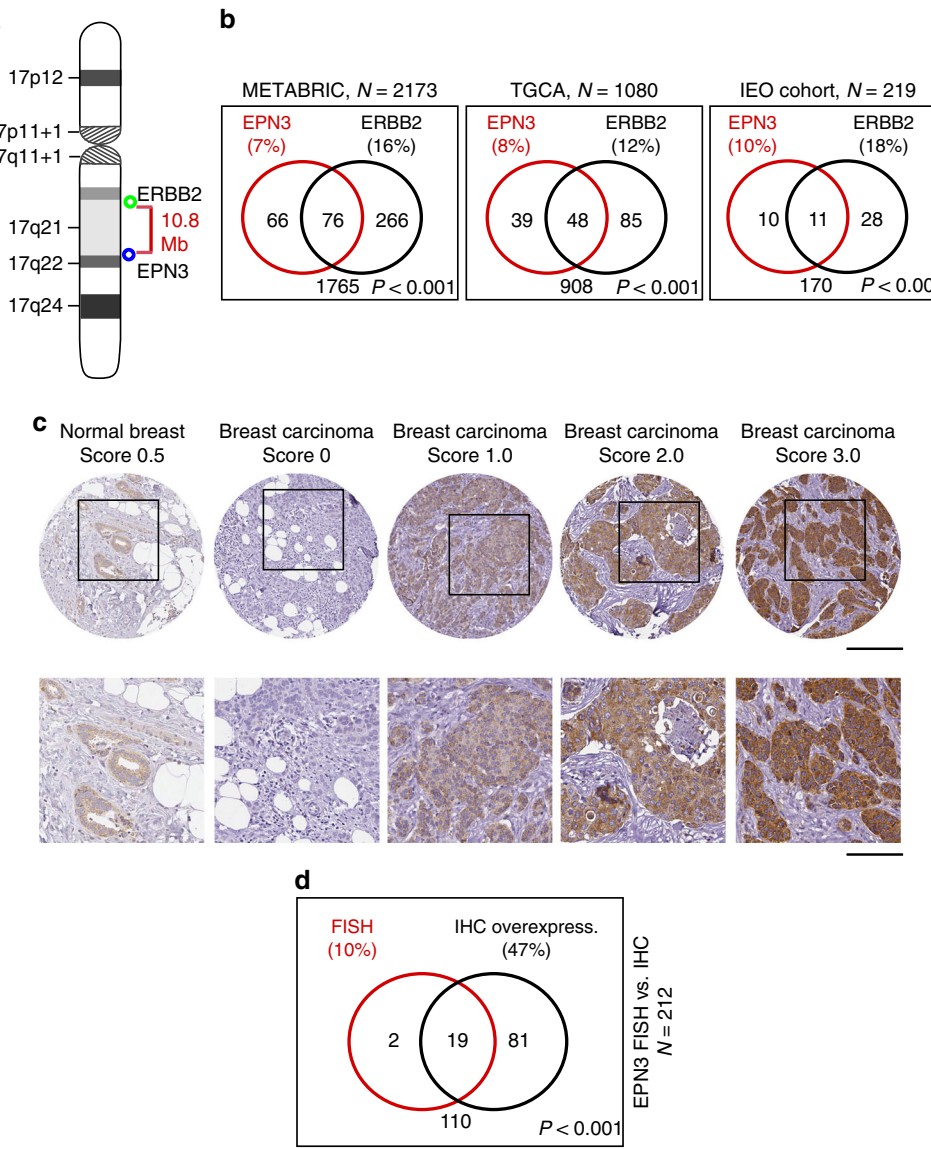

**Fig. 1 Amplification/overexpression of EPN3 in human BCs. a** Schematic representation of human chromosome 17. **b** Venn diagram of *EPN3* and *ERBB2* amplification in different BC cohorts: BC METABRIC cohort ($N = 2137$, available at http://www.cbioportal.org)[71,72], TCGA invasive BC cohort ($N = 1080$, TCGA Research Network: http://cancergenome.nih.gov/), and the IEO cohort ($N = 219$). Percentage (%) of amplification is shown in parentheses. In the IEO cohort, *EPN3* and *ERBB2* were considered amplified when the *EPN3/CEP17* ratio was >2.5, and the *ERBB2/CEP17* ratio was ≥2.0[60], respectively. P, *P* value of the association between the indicated variables by two-sided Fisher's exact test. **c** Representative images of EPN3 IHC (quantification scores are indicated). Top, images at 20× (bar, 200 μm); bottom, magnification of the boxed insets (bar, 200 μm). **d** Venn diagram representation of *EPN3* amplification (FISH) and overexpression (IHC; score >1.0) in the IEO cohort ($N = 212$). P, *P* value of the association between the indicated variables by two-sided Fisher's exact test. Source data are provided as a Source Data file.

showed signs of EMT, with decrease of ECAD expression and acquisition of NCAD staining (Fig. 3b).

**EPN3 drives EMT by stimulating ECAD internalization.** The effects of EPN3 overexpression on ECAD were variable across experiments (compare for instance Figs. 2c, 4f, 5d, 6c, 8g), yet significant (quantification analysis in Fig. 2c and Supplementary Fig. 8C). Furthermore, there was no effect on ECAD mRNA (Fig. 2d). However, changes in ECAD localization or turnover were shown to be critical in inducing partial EMT phenotypes[17,18]. Thus, we analyzed ECAD at cell–cell junctions. In MCF10A-Ctr and MCF10A-EPN1, ECAD staining was uniform in the contact regions (Fig. 4a; Supplementary Fig. 4A). In

MCF10A-EPN3, ECAD signal was lost in a fraction of the cells (Fig. 4a, b). In agreement with what we observed for total ECAD levels, the fraction of cells that lost ECAD expression at the plasma membrane (PM) was variable across experiments (compare standard deviations in Supplementary Fig. 4B right and Fig. 2c), suggesting that cells are in a metastable state in 2D culture, possibly depending on cell confluency and plating conditions. More importantly, in cells that retained ECAD expression, there were marked alterations of the staining pattern, with the formation of comb-like structures with many radially oriented strands (Fig. 4a). While this phenotype has previously been associated with a pre-EMT state[19], it was also markedly different from that in MCF10A-TWIST, in which ECAD staining was evidently decreased (compatible with its transcriptional

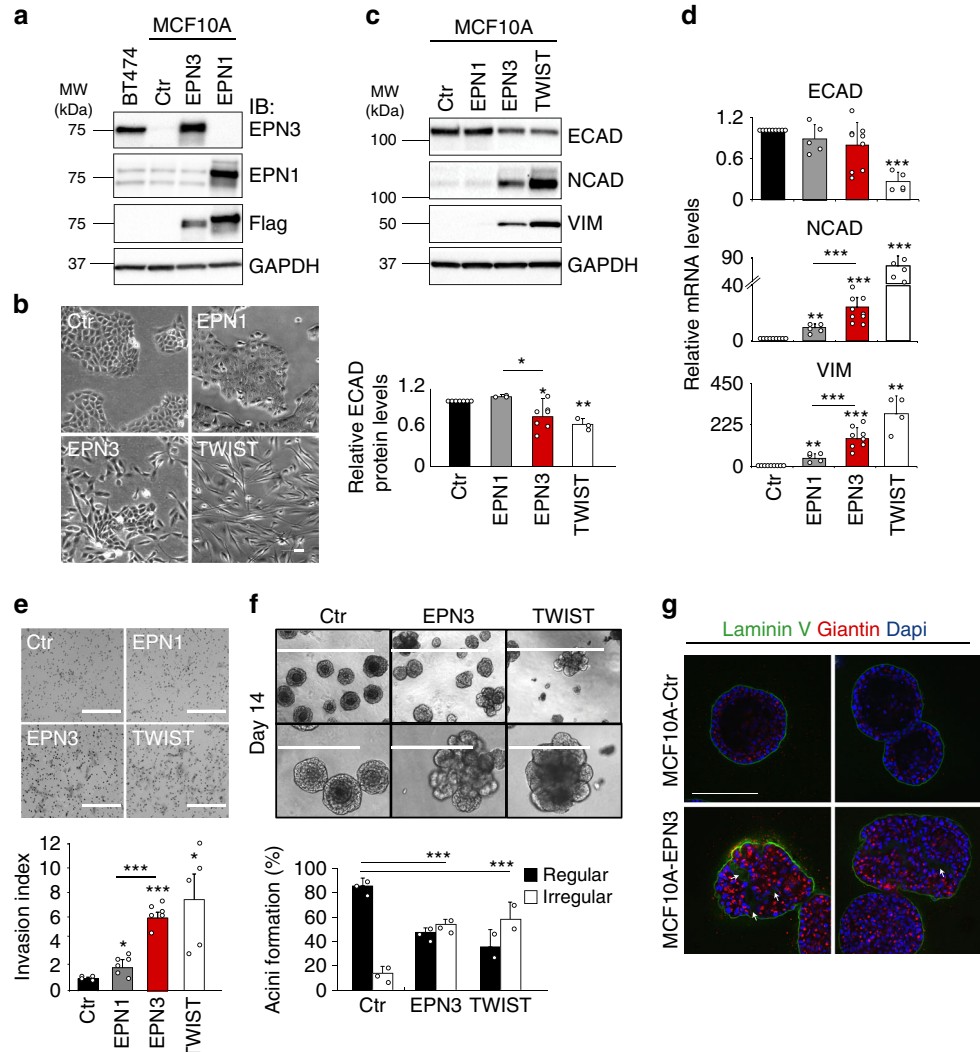

**Fig. 2 EPN3 overexpression induces EMT in MCF10A cells. a** MCF10A cells expressing Flag-EPN3 (EPN3) or Flag-EPN1 (EPN1) or empty vector (Ctr), and IB with the indicated Ab. GAPDH, loading control. BT474 cells: positive control for EPN3 overexpression. Left, MW markers. **b** Phase-contrast microscopy of MCF10A cells as in (**a**), compared with MCF10A-TWIST. Bar, 50 μm. **c, d** Expression of EMT markers, in the indicated cell lines, by IB (**c**) or RT-qPCR (**d**). In (**c**, top), GAPDH, loading control. Left, MW markers. In (**c**, bottom), quantification of ECAD protein levels, normalized on GAPDH expression, is reported as relative fold change compared with Ctr cells. Results are the mean ± S.D. (n ≥ 3). P values (Each Pair Student's t test, two-tailed) are vs. Ctr, unless differently indicated; *, <0.05; **, <0.01; ***, <0.001; ns, not significant (here and in all other figures). In (**d**), the data were normalized on the average expression of the housekeeping genes *18S-ACTB-GAPDH*, and reported as relative fold change compared with Ctr cells (mean ± S.D., n ≥ 4); unless otherwise indicated, the same normalization procedure was used in all figures. **e** Matrigel invasion assay of the indicated cells (24 h). Top, representative images (Bar, 400 μm); bottom, invasion index (mean number of invading cells/field over the control) ± S.D. (at least four fields of view were counted/ sample, each in two independent experiments). P value, Student's t test two-tailed. **f** Morphogenetic Matrigel assay. Top, representative bright-field images at low (bar, 1000 μm) and high magnification (bar, 400 μm) of *acini* grown in Matrigel. Bottom, ratio between regular vs. irregular/multilobular *acini*. Results are reported as mean ± S.D. (n = 3 for Ctr and EPN3; n = 2 for TWIST; at least 75 organoids/experiment were counted); *acini* >50 μm were analyzed for their circularity (regular, circularity >0.8; irregular, circularity <0.8). P value, Fisher's exact t test two-sided for 2 × 2 contingency table. **g** *Acini* as in (**f**), were subjected to IF with the indicated Ab. Blue, DAPI. Two representative images for each sample are shown. Arrows indicate altered basement membrane deposition. Bar, 200 μm. Source data are provided as a Source Data file.

downregulation, Fig. 2d and Supplementary Fig. 4B), but the residual staining appeared morphologically normal (Supplementary Fig. 4A). Also in this case, EPN1 did not phenocopy EPN3 (Supplementary Fig. 4A, B).

These findings suggested that EPN3 overexpression might alter the correct assembly of ECAD at cell–cell contacts. This could be due to altered ECAD endocytosis/trafficking: a process of established relevance in the assembly of adherens junctions and in the establishment of EMT[20]. Indeed, the kinetics of ECAD internalization was significantly accelerated in MCF10A-EPN3 vs. MCF10A-Ctr cells (Fig. 4c; Supplementary Fig. 4C). EPN3 did

not have a general role on endocytosis of other PM receptors (Supplementary Fig. 5A–D), suggesting some degree of specificity.

EPN3 belongs to the epsin family of adaptors that link PM proteins with the endocytic machinery[21]. They also possess lipid-binding and membrane-bending ability[21], and they have been involved in multiple internalization mechanisms, including clathrin-mediated and clathrin-independent mechanisms[22–24]. Both endogenous and overexpressed EPN3 co-immunoprecipitated with components of the ECAD adherens junctions: p120-catenin, β-catenin, and ECAD itself (Fig. 4d). In addition, EPN3

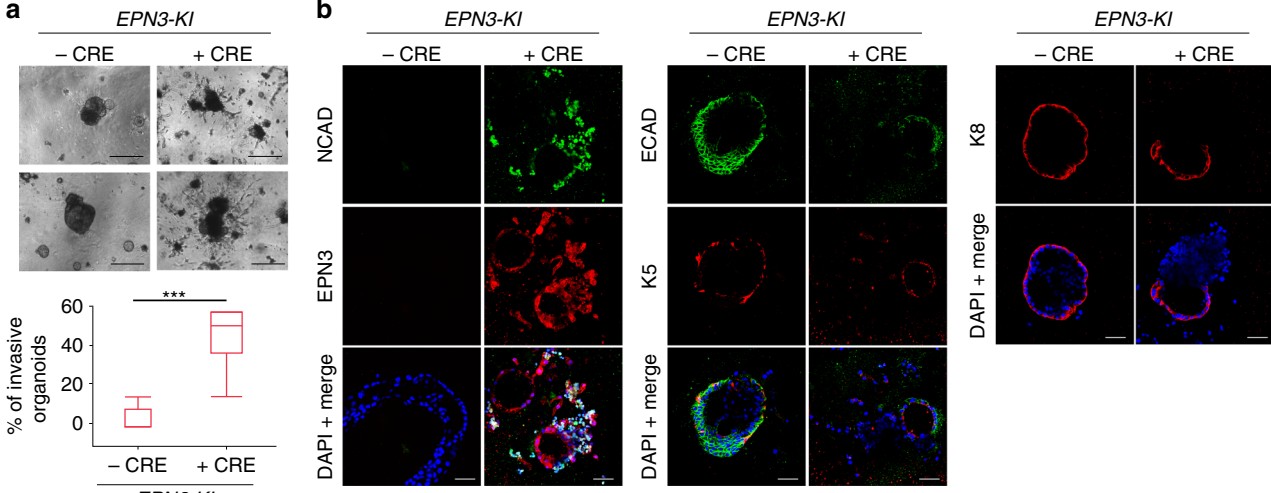

**Fig. 3 EPN3 overexpression induces an invasive phenotype in primary organoids. a** Bilayered primary organoids were obtained from the mammary gland of six *EPN3-KI* mice and treated in vitro (or mock-treated) with Cre. Top, bright-field images. Bar, 270 μm. Bottom, quantification of invasive organoids over total is presented as box plots ($N = 90$ for -Cre and $N = 100$ for +Cre). Here and in other figures, each box plot is defined by 25 and 75 percentiles, showing median, whiskers representing 10 and 90 percentiles and outliers if present. *P* value, Student's *t* test, two-tailed. **b** Bilayered organoids generated as in panel **a** were subjected to IF with Ab against NCAD, ECAD, EPN3, keratin 5 (K5), keratin 8 (K8). Blue, DAPI. Bar, 45 μm. Source data are provided as a Source Data file.

co-immunoprecipitated with clathrin (Fig. 4d). Since ECAD showed some nonspecific binding to control IgGs, we further confirmed the ECAD–EPN3 interaction by in situ proximity ligation assay (PLA) (Supplementary Fig. 5E). These results are compatible with the possibility that EPN3 functions as an endocytic adaptor recruiting the cargo (i.e., ECAD) for internalization. More work will be required, however, to characterize the internalization mechanism of ECAD in these settings.

ECAD loss or its destabilization at cell-to-cell junctions causes the detachment of active (dephosphorylated) β-catenin from the junctions and its nuclear translocation, leading to β-catenin interaction with members of the T cell-specific transcription factor (TCF) family and to the transcriptional activation of EMT genes[25,26]. EPN3-induced ECAD endocytosis correlated with increased nuclear accumulation of active β-catenin in MCF10A-EPN3 *vs.* MCF10A cells (Fig. 4e). Knockdown (KD) of TCF4, a key downstream nuclear β-catenin interactor[27], in MCF10A-EPN3 reverted the EPN3-induced expression of the two EMT targets, VIM and NCAD (Fig. 4f, left). This correlated with the reversion of the elongated, spindle-like, morphology typical of MCF10A-EPN3 (Fig. 4f, right). These results argue that EPN3 overexpression causes destabilization of cell–cell junctions through increased ECAD internalization, followed by β-catenin nuclear translocation and TCF4-dependent upregulation of EMT target genes.

**EPN3 overexpression potentiates TGFβ signaling**. Since the overexpression of EPN3 induces a partial EMT state, arguably through augmented ECAD endocytosis, we investigated whether it might synergize with one of the major inducers of EMT and of ECAD endocytosis, the TGFβ pathway[28–30]. The kinetics of TGFβ-induced ECAD internalization was significantly accelerated in MCF10A-EPN3 vs. MCF10A-Ctr cells (Fig. 5a, b). In addition, EPN3 and ECAD were co-trafficked in the same endocytic structures upon TGFβ stimulation (Fig. 5c), and their interaction by PLA was increased by TGFβ treatment, further supporting their physical interaction (Supplementary Fig. 5E).

We investigated whether EPN3 overexpression cooperates with the TGFβ also in the establishment of EMT phenotypes. We stimulated MCF10A-Ctr and MCF10A-EPN3 with TGFβ at two concentrations (a low suboptimal dose, 0.75 ng ml$^{-1}$, and a high dose, 5 ng ml$^{-1}$) and analyzed early (i.e., SNAIL expression) and delayed (i.e., ECAD, NCAD, VIM and ZEB1 levels) transcriptional responses. In EPN3-overexpressing cells, there was sustained activation of TGFβ signaling at both TGFβ doses, as evidenced by: (i) sustained induction of the transcription factor SNAIL (Fig. 4d, top), (ii) increased basal levels of NCAD and VIM expression both by IB and RT-qPCR analysis, which were further enhanced by TGFβ stimulation (Fig. 5d, top and bottom; Supplementary Fig. 5F, top left), (iii) increased (basal) and sustained (upon stimulation) expression of the late TGFβ-induced transcription factor ZEB1 (Supplementary Fig. 5F, top right) and of FN1 (Supplementary Fig. 5F, bottom left), iv) decreased ECAD protein levels (basal and upon stimulation, Fig. 4d, top). Notably, long-term stimulation (14 days) with TGFβ was able to reduce ECAD levels also at the transcriptional level in MCF10A-EPN3 but not in control cells (Supplementary Fig. 5F, bottom right). Thus, EPN3 overexpression enhances and sustains TGFβ-dependent signaling and transcriptional programs leading to an EMT state, and possibly connected with the acquisition of aggressive phenotypes. Indeed, the combination of suboptimal TGFβ stimulation (incapable of stimulating invasion in control cells) and EPN3 overexpression further enhanced the invasive potential of MCF10A-EPN3 cells in the Matrigel invasion assay (Fig. 5e).

Notably, we confirmed the partial EMT phenotype induced by EPN3 overexpression in a nontumorigenic mammary epithelial cell line of different derivation compared to MCF10A, i.e. MCF12A, and in the non-overexpressing BC cell line HCC1569 (Supplementary Figs. 6 and 7).

Finally, the silencing of endogenous EPN3 in MCF10A reduced both ECAD internalization and transcriptional activation of NCAD and VIM induced by high doses of TGFβ stimulation (Fig. 5f, g), suggesting that EPN3 sustains TGFβ-induced EMT also under physiological conditions, possibly by modulating ECAD endocytosis. These results argue that the effects of EPN3 overexpression represent an exaggeration of its physiological function.

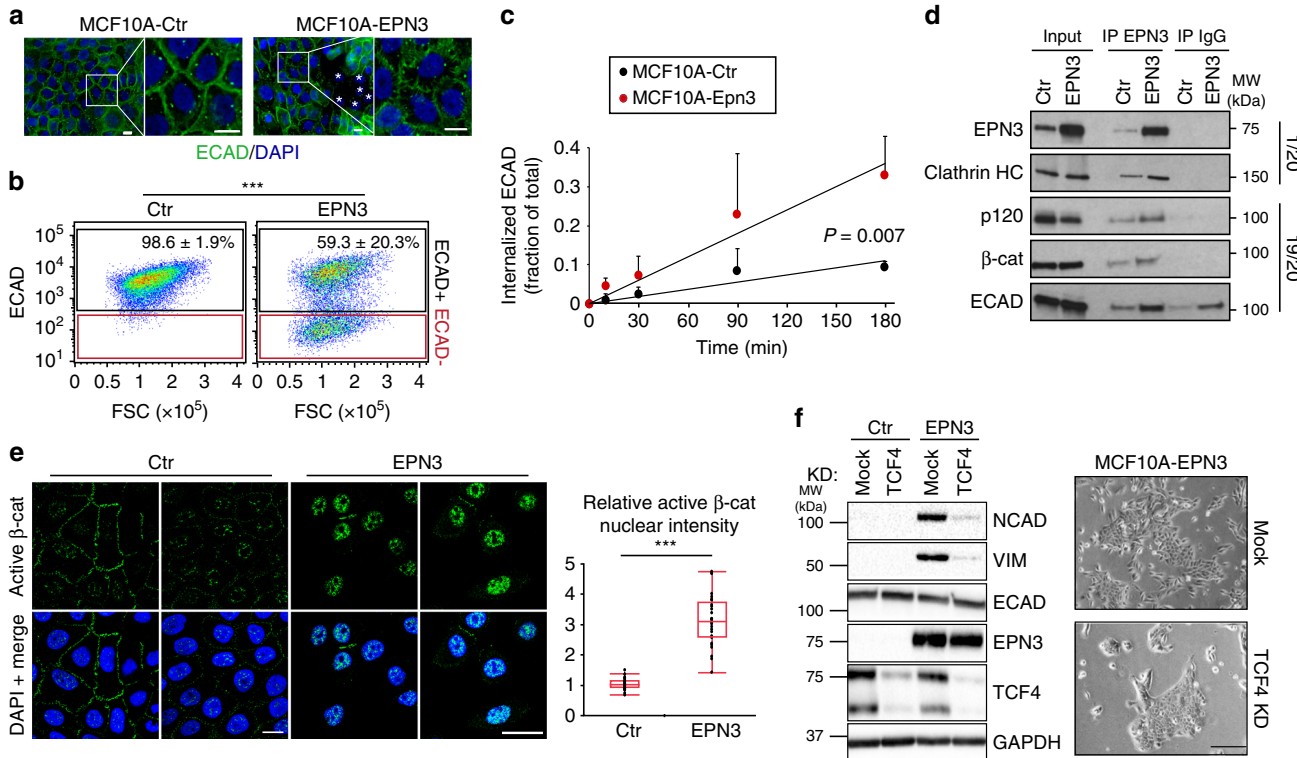

**Fig. 4 EPN3 stimulates ECAD internalization and β-catenin/TCF4-dependent EMT. a** IF staining of ECAD (green) in the indicated cells, at confluency. Blue, DAPI. Asterisks, cells lacking ECAD staining. Bar, 10 μm. **b** Representative FACS analysis of the indicated cells stained in vivo for ECAD. Black and red gates represent ECAD-positive and -negative cells, respectively, and have been set according to the unstained control samples. FSC, forward scatter. Mean percentage of ECAD-positive cells ± S.D. ($n = 18$) is reported. *P* value, Student's *t* test two-tailed. See also Supplementary Fig. 4B. **c** Time course of ECAD internalization in the indicated cells measured by FACS analysis. Fraction of mean fluorescence intensity of internalized ECAD signal over the total ± S.D. ($n \geq 3$) is reported. Student's *t* test, two-tailed. **d** Lysates (1 mg) from the indicated cells were subjected to immunoprecipitation with anti-EPN3 Ab or with a control IgG. Three independent EPN3 IPs were performed to check p120, β-cat, or ECAD binding. Clathrin binding was analyzed in the same IP experiment of β-cat. In each case, efficiency of EPN3 IP gave comparable result (one representative EPN3 IP/IB is shown). In total, 1/20 and 19/20 of the IPs were loaded separately and subjected to IB analysis with the indicated Ab. Input is 20 μg for the 1/20 panel, and 1 μg for the 19/20 panel. Right, MW markers. Results are representative of at least two independent experiments. **e** Left, representative confocal images of the indicated cells stained for active β-catenin. Blue, DAPI. Bar, 30 μm. Right, box plot analyses of relative nuclear fluorescence intensity of active β-catenin per cell in MCF10A-Ctr ($N = 34$) and MCF10A-EPN3 cells ($N = 36$) in a representative experiment ($n = 3$). Data were normalized on the median value of nuclear fluorescence intensity in MCF10A-Ctr. *P* value, each pair, Student's *t* test, two-tailed. **f** TCF4 was silenced in the indicated cells. Negative control, mock transfection. Left, cell lysates were IB with the indicated Ab. GAPDH, loading control. Left, MW markers. See Supplementary Fig. 8C for quantitation and statistics. Right, representative phase-contrast microscopy of mock or TCF4-KD MCF10A-EPN3 cells. Bar, 250 μm. Source data are provided as a Source Data file.

**EPN3-driven partial EMT is due to a TGFβ autocrine loop.** Since EPN3 overexpression renders the cell more responsive to TGFβ, we investigated whether the endogenous TGFβ signaling pathway plays a role in the EPN3-induced partial EMT state. In EPN3-overexpressing cells, the transcript levels of TGFβ ligand 1 (TGFβ1), ligand 2 (TGFβ2), and receptor 2 (TGFβR2) were significantly increased vs. control cells (Fig. 6a). Moreover, increased TGFβ1 was detected in the supernatant of EPN3-overexpressing cells vs. MCF10A (Fig. 6b) or MCF10A-EPN1 cells (Supplementary Fig. 8A). This suggested the existence of a transcriptional positive feedback, leading to the creation of a TGFβ-dependent autocrine loop, required to sustain the partial EMT phenotype in EPN3-overexpressing cells. To test this possibility, we silenced TGFβ receptors (type 1 and 2, individually or in combination) in MCF10A-EPN3 cells (Fig. 6c; Supplementary Fig. 8B) and observed reversion of EMT evidenced by: (i) reduction in NCAD and VIM proteins (Fig. 6c, top; Supplementary Figs. 8B, right and 8C) and mRNA levels of TGFβ1, NCAD and VIM (Fig. 6d; see also Supplementary Fig. 8D), and (ii) reversion of the fibroblast-like morphology toward an epithelioid one (Fig. 6c, bottom). EPN3 overexpression and TGFβ stimulation acted synergistically

not only on NCAD and VIM mRNA levels but also on TGFβ1 mRNA (Fig. 6d), reinforcing the notion of the existence of a feedback loop that acts on EMT and on TGFβ signaling itself. These results were confirmed using a chemical inhibitor of the kinase activity of TGFβR1-2, LY2109761, which was able to revert the EPN3-dependent phenotypes (Supplementary Fig. 9A, B)[31].

The EMT reversion upon TGFβR1 ablation and LY2109761 treatment was comparable with that achieved by silencing TCF4 (Fig. 6c, d), which could also act downstream of TGFβ signaling[32]. Consistently, the EMT reversion upon TCF4 KD was observable in MCF10A-EPN3 cells both in basal and TGFβ-stimulated conditions (Fig. 6d). This result shows that the EPN3-induced autocrine TGFβ-based loop is TCF4-dependent.

Finally, we note that the reversion of EPN3-induced partial EMT upon TGFβR KD, TCF4 KD, or LY2109761 treatment was not complete. Although we cannot exclude that this is due to incomplete KD or inhibition, these data seem to argue that EPN3-induced partial EMT is—to some extent—also independent of TGFβ signaling and/or TCF4 activity, an issue that will require further investigations.

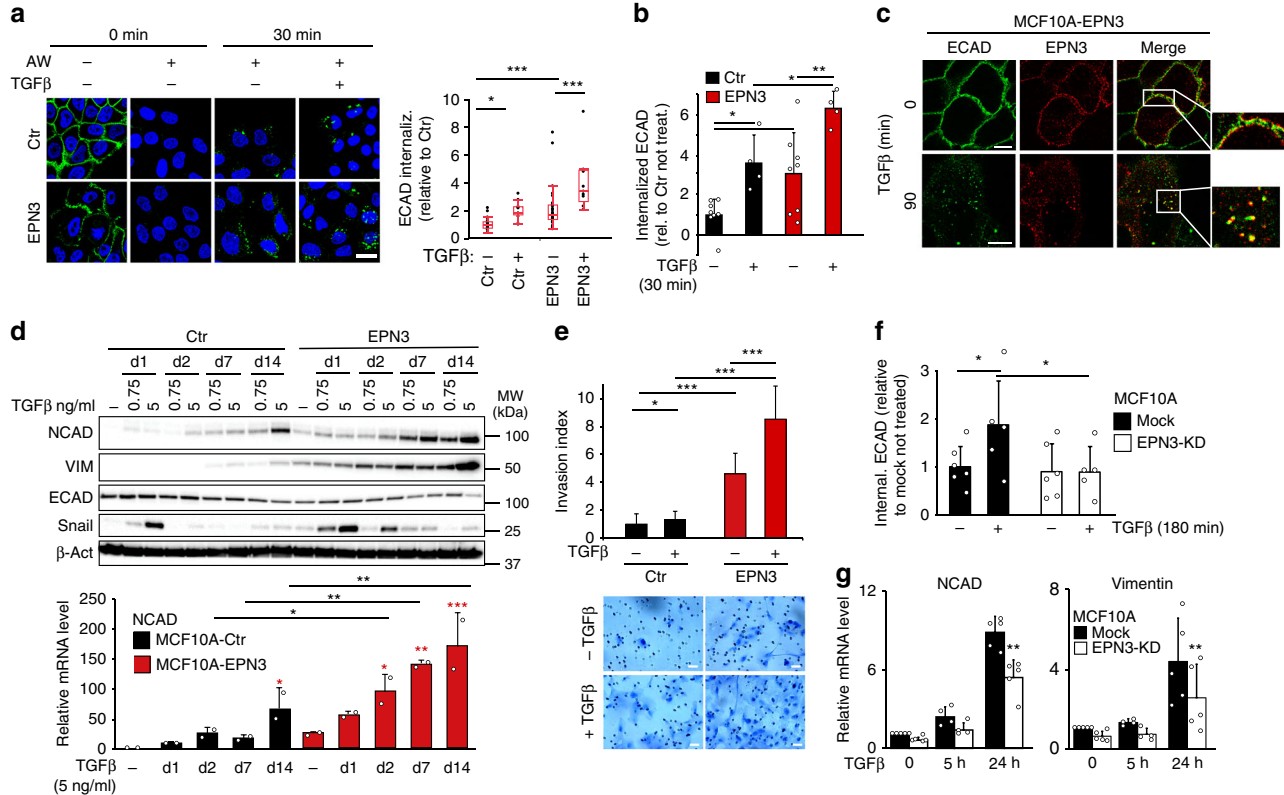

**Fig. 5 EPN3 is involved in TGFβ-induced ECAD internalization and TGFβ signaling. a** Left, ECAD internalization assay by IF in the indicated cells using an anti-ECAD Ab against the extracellular domain internalized for 30 min −/+ TGFβ (5 ng ml$^{-1}$). Cells were subjected or not to an acid wash (AW) to remove the ECAD Ab from the PM, prior fixation. Bar, 20 μm. Right, box plot of relative ECAD fluorescence intensity/field of view ($N \geq 9$). P value, each pair Student's t test two-tailed. **b** ECAD internalization assay by FACS in the indicated cells, −/+ TGFβ1 (5 ng ml$^{-1}$, 30 min). Mean fluorescence intensity of internalized ECAD/total ± S.D. ($n \geq 4$) is reported. P value, Student's t test two-tailed. **c** Representative co-localization images of EPN3 (red) and ECAD (green), during ECAD internalization as in (**a**). Bars, 10 μm. **d** Top, lysates from the indicated cells −/+ TGFβ1 (0.75 or 5 ng ml$^{-1}$, for the indicated days—d) were IB with the indicated Ab. β-actin, loading control. Right, MW markers. Bottom, NCAD mRNA levels by RT-qPCR on samples −/+ TGFβ1 (5 ng ml$^{-1}$). Mean relative mRNA expression vs. MCF10A-Ctr (not stimulated) ± S.D. (two independent experiments, each in technical triplicates) is reported. P value, each pair Student's t test, two-tailed. Black asterisks, statistical significance between each corresponding time point of EPN3 vs. Ctr; red asterisks, statistical significance between stimulated vs. not stimulated samples of MCF10A-EPN3 or -Ctr cells. **e** Matrigel invasion assay of the indicated cells −/+ TGFβ (0.75 ng ml$^{-1}$). Top, invasion index (mean number of invading cells/field vs. control) ± S.D. ($N \geq 22$). P value, Student's t test two-tailed. Bottom, representative images. Bar, 50 μm. **f** ECAD internalization assay by FACS analysis in MCF10A-Ctr −/+ EPN3 silencing, −/+ TGFβ1 (5 ng ml$^{-1}$, 180 min). Mean fluorescence intensity of internalized ECAD/total ± S.D. ($n \geq 5$) is reported. P value, Student's t test two-tailed. **g** RT-qPCR analysis of NCAD and VIM mRNA levels in cells as in (**f**). Mean relative mRNA expression vs. mock-unstimulated cells ± S.D. ($n \geq 4$) is reported. P value, each pair Student's t test, two-tailed. Source data are provided as a Source Data file.

**ECAD endocytosis is the first event in EPN3-driven EMT.** Together, our data suggest a scenario in which accelerated ECAD endocytosis, induced by EPN3 overexpression, activates the β-catenin/TCF4-dependent EMT program, as well as a TGFβ-dependent loop that positively feedbacks on ECAD endocytosis, on TCF4-dependent EMT and on TGFβ signaling itself, thereby establishing the partial EMT phenotype and the invasive properties of the cells (Fig. 6e).

To challenge this model and, in particular, to test whether ECAD endocytosis is the initial event in EPN3-driven partial EMT, we generated a doxycycline-inducible EPN3 overexpression system in MCF10A cells (Fig. 7a). This system allowed us to define the temporal order of the events. At 6 h after induction, we observed a slight change in cell morphology, which became more dramatic after 24 h (Fig. 7b). Importantly, TGFβ ligands and receptors, as well as other EMT factors, were not yet upregulated at 6 h of induction (Fig. 7c; Supplementary Fig. 9C). Despite the lack of TGFβ pathway activation, ECAD endocytosis was accelerated (Fig. 7d, e), cell-to-cell junctions were rearranged (Fig. 7d) and active β-catenin was translocated to the nucleus

(Fig. 7f). These results demonstrate that these events occur before the establishment of the TGFβ-dependent loop and most likely represent the first steps of EPN3-induced partial EMT.

Notably, once the loop is established, the β-catenin/TCF4 system likely cooperates with TGFβ signaling, as supported by the evidence that the TGFβ-dependent transcriptional events are also in part dependent on β-catenin/TCF4 (Fig. 6d; Supplementary Fig. 8D).

Finally, by taking advantage of the inducible system, we could recapitulate some of the EPN3-dependent phenotypes also in MCF10A-derived organoids grown in 3D Matrigel where, by inducing the overexpression of EPN3 at the end of the morphogenesis process, we observed rearrangements of ECAD cell-to-cell junctions and the translocation of active β-catenin into the nucleus (Fig. 7g).

**EPN3 overexpression causes the appearance of CSC traits.** EMT has been linked to the emergence of CSCs[4]. We searched for the appearance of CSC traits in EPN3-overexpressing cells by exploiting the CD44$^{high}$/CD24$^{low}$ configuration, which has been

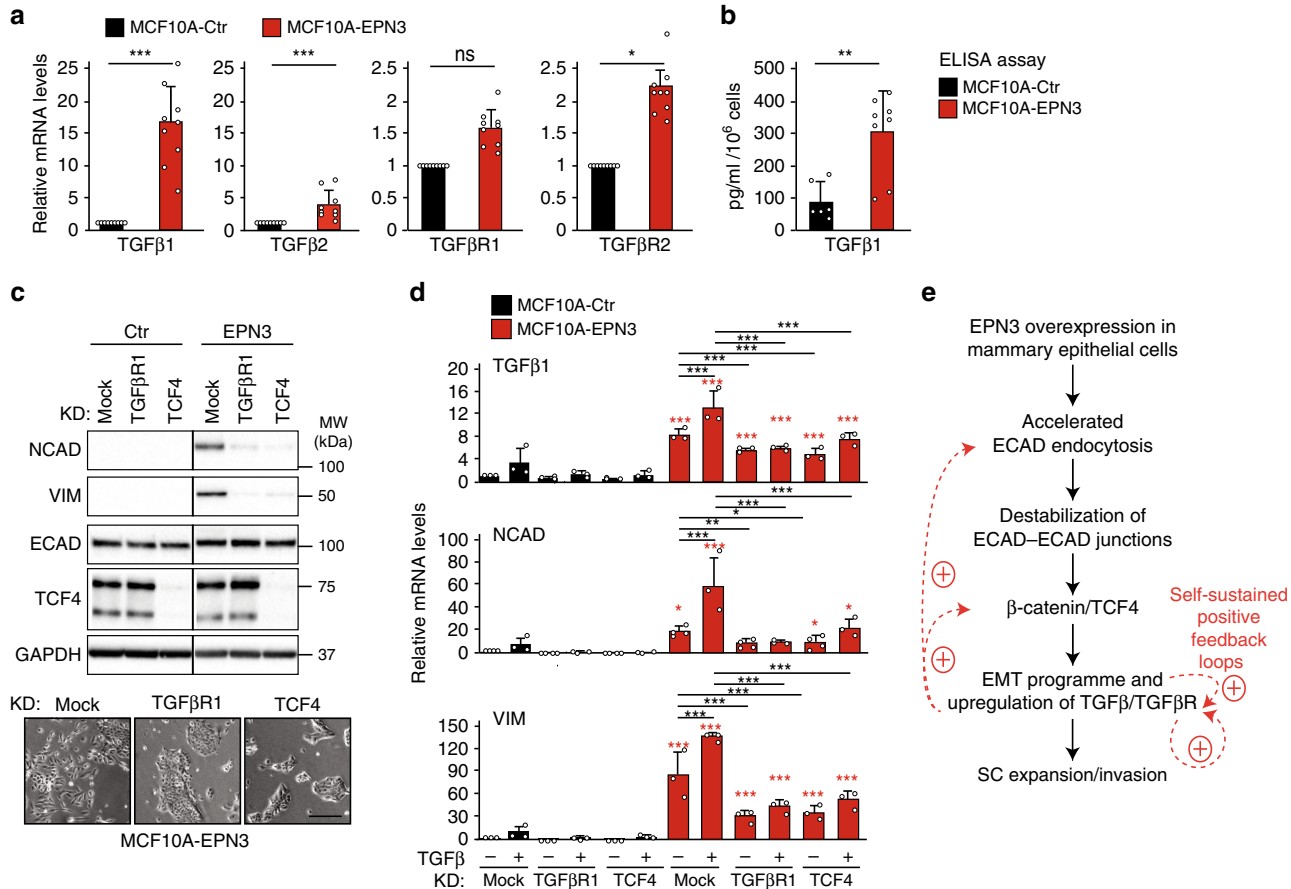

**Fig. 6 EPN3-driven partial EMT relies on a TGFβ-based autocrine loop. a** RT-qPCR analysis of mRNA expression of TGFβ ligands (TGFβ1, TGFβ2) and receptors (TGFβR1, TGFβR2) in MCF10A-Ctr and -EPN3 cells. Mean relative mRNA expression compared with MCF10A-Ctr ± S.D. of nine independent experiments, each performed in technical triplicates. *P* value, Student's *t* test two-tailed. **b** ELISA assay of secreted TGFβ1 in conditioned media from MCF10A-Ctr and -EPN3 cell cultures. Results are reported as pg ml$^{-1}$ 10$^{-6}$ cells ± S.D. (*n* = 3, each performed on the medium of at least two independent dishes, each assayed in technical triplicates). *P* value, Student's *t* test two-tailed. The same data are also reported in Supplementary Fig. 8A top for comparison between MCF10A-Ctr, -EPN3, -EPN1, and -TWIST. **c** Top, TCF4 or TGFβR1 were silenced in the indicated cells. Mock transfection was used as negative control. Cell lysates were IB with the indicated Ab. This panel was assembled from samples run on the same gel by splicing out the irrelevant lanes (as shown by the black lines). GAPDH, loading control. Right, MW markers. Relative protein levels of NCAD, ECAD and VIM of this blot were quantified together with other experiments (see Supplementary Fig. 8C for quantitation and statistics). Bottom, representative phase-contrast microscopy of MCF10A-EPN3 cells, mock, TCF4 KD or TGFβR1 KD. Bar, 250 μm. **d** RT-qPCR analysis of mRNA expression of the indicated genes in MCF10A-Ctr and -EPN3 cells that were mock-treated or subjected to TGFβR1 KD or TCF4 KD, and stimulated or not with TGFβ1 (5 ng ml$^{-1}$, 48 h). Results are reported as relative mRNA expression compared with the MCF10A-Ctr mock-treated sample, mean ± S.D. (*n* ≥ 3, each performed in technical triplicates). *P* value, each pair Student's *t* test, two-tailed. Black asterisks, statistical significance between KD samples vs. mock, not treated, and KD samples vs. mock, treated with TGFβ1, as indicated by the bar; red asterisks, statistical significance between matched MCF10A-EPN3 samples vs. -Ctr samples. **e** Proposed model of the EPN3-dependent activation of the EMT program in mammary epithelial cells. Source data are provided as a Source Data file.

associated with EMT and human mammary CSCs[4,33]. MCF10A-Ctr or -EPN1 cells displayed a very small percentage of these cells (0.1% and 0.4%, respectively); conversely MCF10A-EPN3 or -TWIST cells contained ~7% and 30% of this subpopulation, respectively (Fig. 8a, b; Supplementary Fig. 10A).

At the biological level, we employed the mammosphere (MS) assay, which can be utilized (with due caution) as a proxy for the detection of cells with stem-like properties[34,35]. In this assay, MCF10A-EPN3, similarly to MCF10A-TWIST, displayed a sphere-forming efficiency (SFE) at least tenfold higher than control or EPN1-overexpressing cells (Fig. 8c). The increase in SFE in MCF10A-EPN3 vs. -Ctr (or -EPN1) significantly correlated with the increase in the CD44$^{high}$/CD24$^{low}$ population (compare Fig. 8c, b). In agreement, the MS-forming ability resided almost exclusively in the CD44$^{high}$/CD24$^{low}$ subpopulation (Fig. 8d), which is also the subpopulation displaying EMT

traits, as confirmed by the expression pattern of EMT-specific markers (Fig. 8e).

Finally, the percentage of CD44$^{high}$/CD24$^{low}$ CSC-like cells in MCF10A-EPN3 (or MCF10A-TWIST) was considerably reduced by TGFβR1 KD (Fig. 8f), which correlated with reversion of EMT (Fig. 8g).

**The transition from in situ to invasive BC involves EPN3.** Augmented invasiveness in vivo is connected with the acquisition of EMT traits and CSC-like properties[4,5]. To investigate this possibility, we took advantage of the human BC cell line MCF10DCIS.com, a derivative of MCF10A cells, which reproducibly forms comedo DCIS (ductal carcinoma in situ)-like lesions that spontaneously progress to invasive tumors in xenograft models[36].

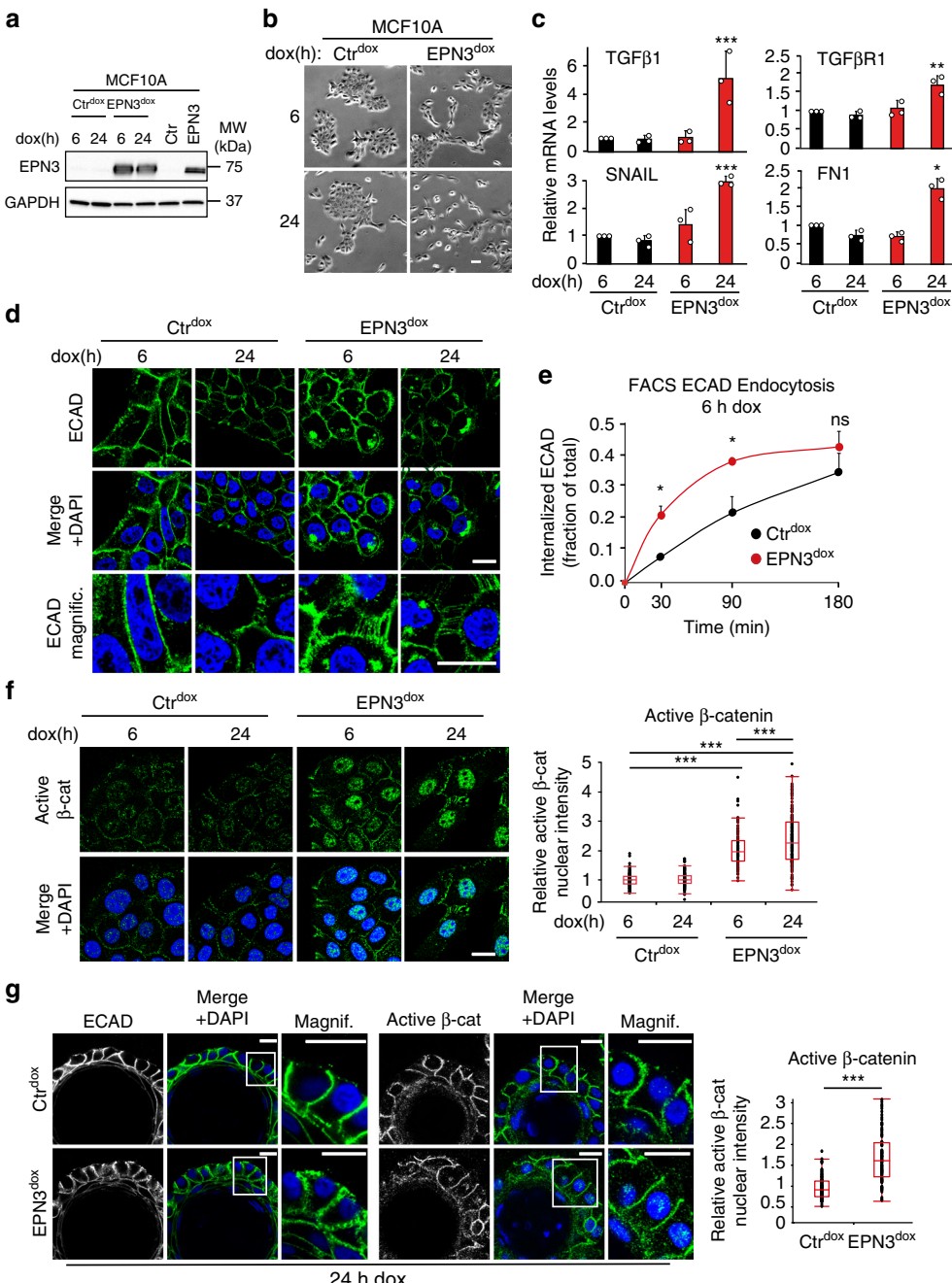

**Fig. 7 ECAD accelerated endocytosis is the initial event in EPN3-driven EMT. a** MCF10A-Ctr cells were infected with pSLIK-empty vector (Ctr[dox]) or pSLIK-EPN3 (EPN3[dox]) and treated for 6 h or 24 h with 200 µg ml[−1] doxycycline (dox). Lysates were subjected to IB analysis with the indicated Ab. GAPDH, loading controls. Lysates of MCF10A-Ctr and MCF10A-EPN3 were loaded in parallel as controls. Right, MW markers. **b** Phase-contrast microscopy of cells as in (**a**). Bar, 60 µm. **c** RT-qPCR analysis of mRNA expression of the indicated genes in cells as in (**a**). Mean relative mRNA expression compared with MCF10A-Ctr, ± S.D. is reported (*n* = 3, each performed in technical triplicates). *P* value, each pair Student's *t* test, two-tailed. **d** IF staining of ECAD (green) in cells (**a**). Blue, DAPI. Bar, 20 µm. **e** Time course of ECAD internalization measured by FACS analysis in MCF10A-Ctr[dox] and -EPN3[dox] cells treated with dox for 6 h. Data are reported as fraction of mean fluorescence intensity of internalized ECAD signal over the total, ± S.D. (*n* = 3). *P* values, Student's *t* test two-tailed. **f** Left, representative confocal images of the indicated cells dox-induced for 6 h and 24 h, and stained for active β-catenin. Blue, DAPI. Bar, 20 µm. Right, box plot analyses of relative nuclear fluorescence intensity of active β-catenin per cell in MCF10A-Ctr[dox] 6 h (*N* = 222), MCF10A-Ctr[dox] 24 h (N = 323), MCF10A-EPN3[dox] 6 h (*N* = 98) and MCF10A- EPN3[dox] 24 h cells (*N* = 144) in a representative experiment (*n* = 3). Data were normalized on the median value of nuclear fluorescence intensity in the MCF10A-Ctr[dox] 6 h sample. *P* value, each pair Student's *t* test, two-tailed. **g** Left, *acini* grown in 3D on top of Matrigel for 13 days were treated for 24 h with doxycycline and then subjected to IF with the indicated Ab. Blue, DAPI. Bar, 20 µm. Right, box plot of relative nuclear fluorescence intensity of active β-catenin per cell in a representative experiments of two independent replicates. Data were normalized on the median value of nuclear fluorescence intensity in the MCF10A-Ctr[dox] 24 h sample. MCF10A- Ctr[dox] (*N* = 86) and MCF10A-EPN3[dox] cells (*N* = 91). *P* value, each pair Student's *t* test two-tailed. Source data are provided as a Source Data file.

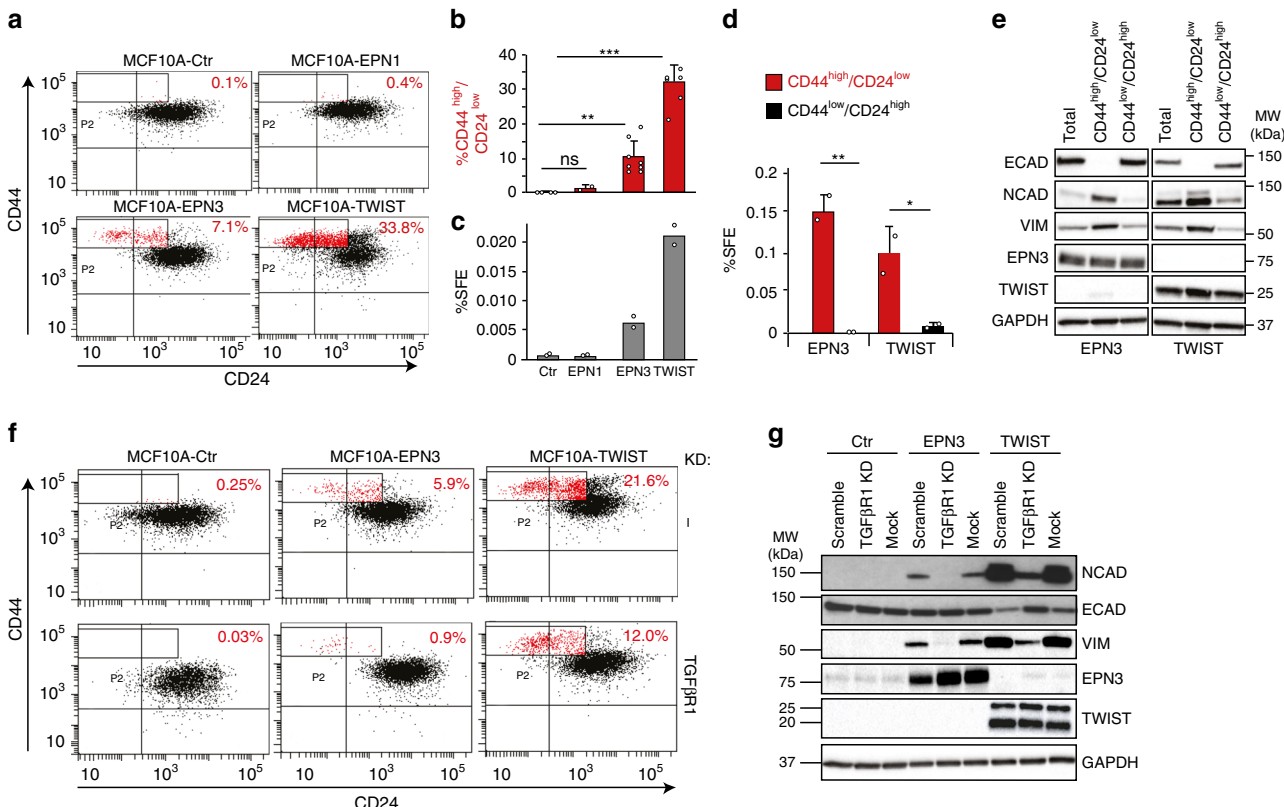

**Fig. 8 EPN3 overexpression expands a cellular compartment with features of CSCs. a** Representative bi-parametric (CD44/CD24) FACS analysis of MCF10A cells expressing the indicated constructs. The CD44$^{high}$/CD24$^{low}$ population is boxed and shown in red, with the relative percentage. See also Supplementary Fig. 10A for a different representation of the results. **b** The size of the CD44$^{high}$/CD24$^{low}$ population (measured by FACS as in **a**) in the indicated samples is reported as a percent of the total. Mean ± S.D. of at least two independent experiments is shown (N, Ctr=4, EPN1 = 2, EPN3 = 8, TWIST = 6). P value, each pair Student's t test two-tailed. **c** Mammosphere assay on MCF10A cells expressing the indicated constructs. Results are expressed as mean sphere-forming efficiency (SFE) of two independent experiments. P value, each pair Student's t test two-tailed. **d** Mammosphere assay on the CD44$^{high}$/CD24$^{low}$ and CD44$^{low}$/CD24$^{high}$ FACS-sorted subpopulations of MCF10A cells expressing the indicated constructs. Results are expressed as mean SFE ± S.D. of two independent experiments, each performed in technical triplicates. P value, each pair Student's t test two-tailed. **e** The indicated FACS-sorted populations from MCF10A-EPN3 or -TWIST cells were analyzed by IB as shown and compared to the total unsorted population. GAPDH, loading control. This panel was assembled from samples run on the same gel by splicing out the irrelevant lanes, as shown by the black lines. MW markers are shown on the right. **f** Representative bi-parametric (CD44/CD24) FACS analysis of the indicated cell lines, mock-silenced (Ctr, top) or silenced for TGFβR1 (bottom). The CD44$^{high}$/CD24$^{low}$ population is boxed and shown in red, with the relative percentage. Results are representative of two independent experiments. **g** MCF10A cells expressing the indicated constructs were transfected with a non-targeting oligo (scramble), mock-transfected (mock), or silenced for TGFβR1 (TGFβR1 KD) and analyzed by IB. GAPDH, loading control. MW markers are shown on the left. Source data are provided as a Source Data file.

By analyzing the expression levels of ECAD, VIM, and NCAD, we showed that MCF10DCIS.com cells, grown in 2D, exhibited some traits of EMT, as indicated by the expression of NCAD and VIM (Fig. 9a). The KD of EPN3 caused a partial reversion of this EMT phenotype in 2D culture conditions, as witnessed by a decrease in VIM expression and slight upregulation of ECAD (Fig. 9a). When engrafted into the mammary fat pad, tumors formed by parental MCF10DCIS.com recapitulated the progression of DCIS into invasive BC (IBC), as confirmed by IHC analysis using the basal-myoepithelial marker p63 (Fig. 9b), that has been shown to increase during the progression from DCIS to IBC[36]. Moreover, in DCIS, p63 marked the myoepithelial cell layer, while in IBC the majority of infiltrating cells became p63-positive, accompanied by a concomitant increase in α-SMA-positive fibroblasts in the stroma associated with IBC, as previously reported (Fig. 9b, c)[36]. In addition, cells in the invasive area lost the epithelial marker ECAD and acquired the mesenchymal marker NCAD, thus recapitulating EMT in vivo (Fig. 9c, red arrowheads). Concomitantly, endogenous EPN3 levels were significantly increased in the infiltrating areas (Fig. 9c,

d, red vs. black arrowheads), suggesting a possible role of EPN3 in the progression from in situ to IBC. Indeed, EPN3 silencing in MCF10DCIS.com cells not only reduced tumor growth (Supplementary Fig. 10B), but importantly, also reduced the p63/α-SMA-positive area of tumors during IBC progression over time, when normalized on tumor size (Fig. 9e, f).

These data indicate that, in the in vivo 3D context, the effects of EPN3 silencing are more pronounced than in 2D (Fig. 9a), causing a decrease in the infiltrating tumor areas that have undergone EMT in vivo (Fig. 9f; Supplementary Fig. 10C). Thus, EPN3 upregulation participates to the progression from in situ to invasive BC.

**High EPN3 level is an independent predictor of BC metastasis.** To investigate the relevance of this finding to human BCs, we selected a subset of human primary BCs (N = 50), in which we could detect both DCIS and IBC areas in the same slide. EPN3 levels were upregulated in the infiltrating area adjacent to the in situ component in ~20% of these samples (11 out of 50, Fig. 10a, b, red dots), showing that EPN3 endogenous levels are

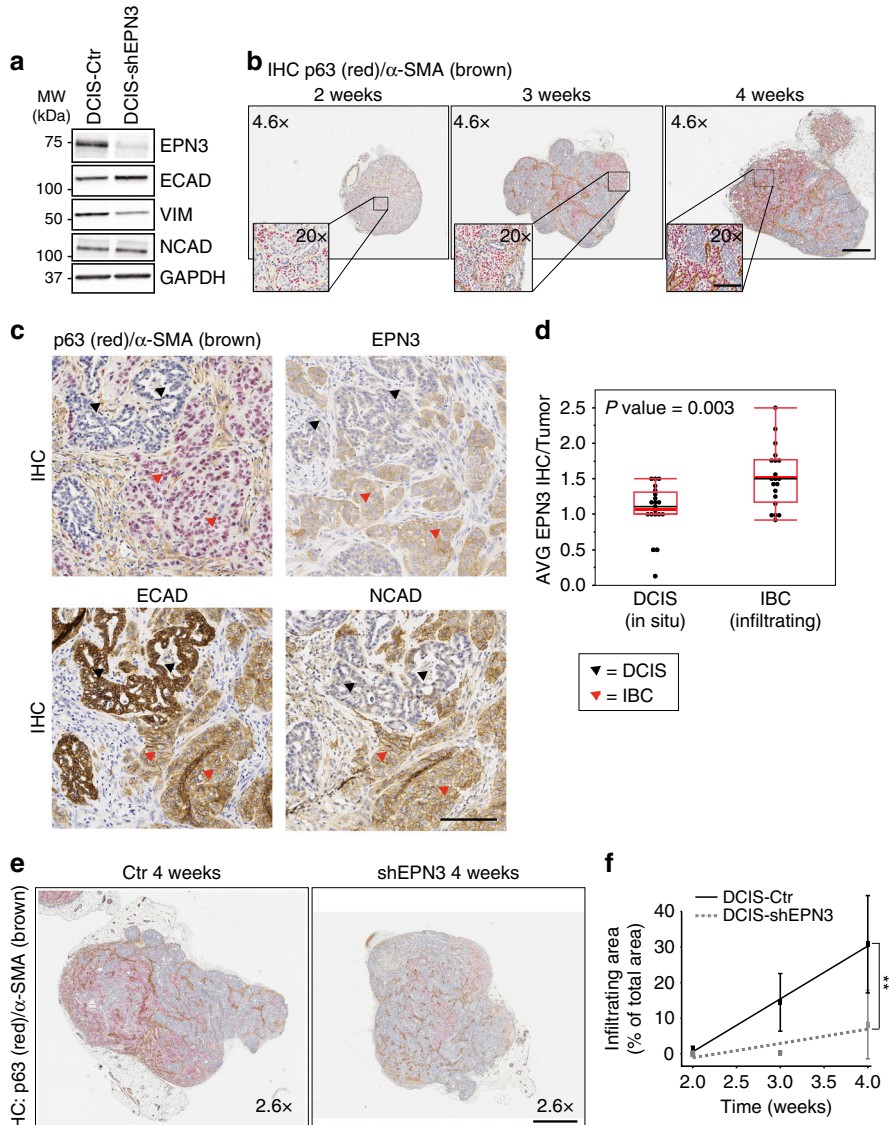

**Fig. 9 EPN3 expression during the progression from in situ to invasive BC. a** Expression of EMT markers in MCF10DCIS.com cells infected with control vector (DCIS-Ctr) or shEPN3 (DCIS-shEPN3) by IB with the indicated Ab. GAPDH, loading control. Left, MW markers. **b** IHC double staining of p63 (red)/α-SMA (brown) in tumors grown from MCF10DCIS.com cells engrafted in the mammary fat pad over time (2–3–4 weeks). Bar, 500 μm. Inset, magnifications (×20) of the indicated areas. Bar, 100 μm. **c, d** Five consecutive sections of 20 DCIS-Ctr tumors derived as described in (**a**) and (**b**) were stained with H&E (not shown), p63/α-SMA, EPN3, NCAD, or ECAD Abs. **c** IHC images of in situ (DCIS, black arrowheads) and infiltrating (IBC, red arrowheads) areas of a representative DCIS-Ctr tumor stained with the indicated Ab on consecutive sections. p63 (red) and α-SMA (brown) were double stained. Images at ×20. Bar, 100 μm. **d** All the areas with clearly distinct components of DCIS present in each tumor, and adjacent IBC areas, were identified based on H&E and p63/α-SMA staining. In these matched DCIS vs. IBC areas, EPN3 expression was evaluated by IHC using intensity scores ranging from 0 to 3 as described for TMA analysis (see "Methods"), and an average score was then calculated for all the DCIS and IBC areas identified within each tumor. Box plot of the average EPN3 expression measured in DCIS and paired IBC areas of each MCF10DCIS.com-Ctr tumor at 2, 3, 4, 5 weeks (N, number of tumors analyzed = 20). P value, Wilcoxon test. **e** Representative images of the double IHC staining of p63 (red)/α-SMA (brown), in DCIS-Ctr and DCIS-shEPN3 tumors at 4 weeks. One image for each tumor is shown (×2.6). Bar, 800 μm. The red/brown color marks the invasive areas. **f** Percentage of infiltrating areas normalized on the total tumor area was measured for each DCIS-Ctr and DCIS-shEPN3 tumor at each time point. Results at each time point represent the mean ± S.D. of at least four tumors per time point per condition (DCIS-Ctr and -shEPN3). P value, Student's t test two-tailed. Source data are provided as a Source Data file.

upregulated in a fraction of human BCs that are undergoing the in situ to invasive transition.

Given the link between EPN3 and the invasive behavior of cancer cells, we aimed to establish the clinical relevance of EPN3 to BC metastasis. By analyzing EPN3 expression (IHC or RT-qPCR) in a large retrospective consecutive cohort (IEO BC 97-00) of ~2400 BCs (Supplementary Tables 1–2)[14], we determined that high EPN3 expression was an independent predictor of poor prognosis, in multivariable analysis, associated with a higher risk of distant metastasis (Fig. 10c; Supplementary Table 3). The prognostic value of a high EPN3 status was linked to the metastatic phenotype per se, as no correlations were found with loco-regional disease recurrence (Fig. 10c; Supplementary Table 3). Importantly, EPN3 overexpression was an independent predictor of unfavorable outcome in node-negative patients (by RT-qPCR) and in the subgroup of ERBB2-negative

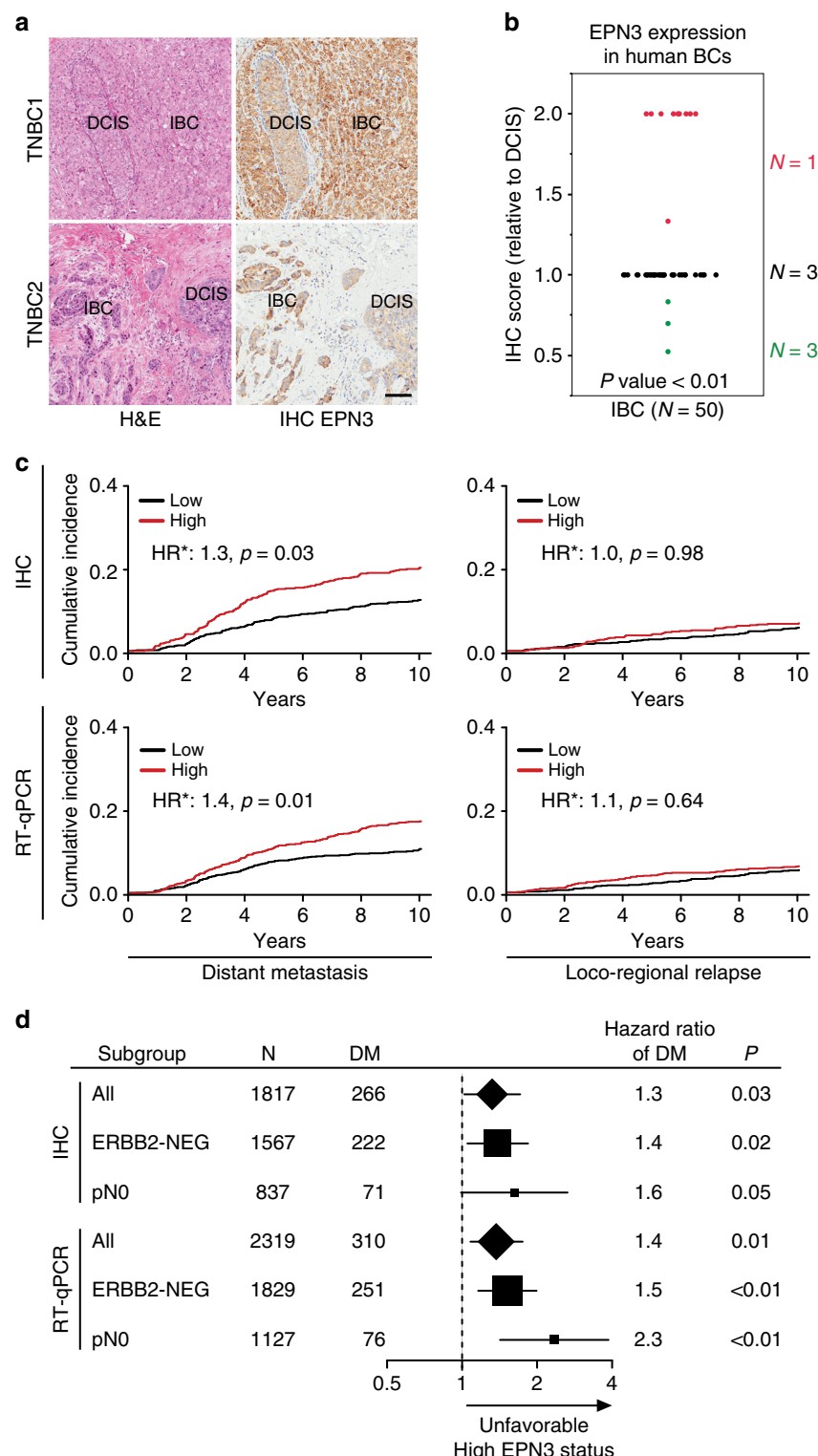

patients (by RT-qPCR and IHC, Fig. 10d; Supplementary Table 3).

## Discussion

We herein demonstrate that EPN3 is a bona fide oncogene in BC, as supported by: (i) genetic alterations of the *EPN3 locus*; (ii) high frequency of EPN3 overexpression, in the presence or absence of gene amplification; (iii) its ability to independently predict metastasis; (iv) the attenuation upon EPN3-KD, and the increase upon EPN3 overexpression, of in vivo tumorigenicity of BC cell lines harboring or not *EPN3* gene amplification, respectively; (v) the induction of EMT, and the appearance of cells with traits of CSCs, upon EPN3 overexpression in model systems in vitro; (vi) the invasive phenotype induced by EPN3 overexpression in ex vivo primary mammary organoids; (vii) the upregulation of EPN3 at the leading edge of infiltrating human BCs; (viii) the attenuation upon EPN3-KD of the transition from in situ to invasive carcinomas in vivo.

**Fig. 10 High EPN3 levels correlate with IBC and independently predict metastasis. a** H&E staining and EPN3 IHC images of in situ (DCIS) and adjacent infiltrating area (IBC) detected in two different primary human BCs. Images at ×20. Bar, 100 μm. **b** Summary graph of the differential expression of EPN3 detected by IHC in human BC samples in areas of DCIS vs. adjacent IBC. Data are reported as IHC score in IBC relative to DCIS. *N*, number of tumor samples. *P* value, two-sided Wilcoxon signed-rank test. **c** Cumulative incidence of distant metastasis or loco-regional relapse in BC patients of the IEO consecutive cohort (IEO BC 97-00, *N* = 2453), stratified by EPN3 protein (top, IHC) or mRNA (bottom, RT-qPCR) levels. HR*, multivariable hazard ratio. Multivariable models were adjusted for tumor grade, tumor size, Ki-67 levels, ERBB2 status, ER/PR status, number of positive lymph nodes, and age at surgery, see Supplementary Table 3. **d** Forest plot of the multivariable hazard ratios of distant metastasis and 95% Wald confidence intervals (whiskers) in the entire cohort of patients (All) and in the ERBB2-negative (ERBB2-NEG) or lymph node-negative (pN0) subgroups of patients stratified by EPN3 protein (IHC) or mRNA (RT-qPCR) expression levels. The size of the solid squares and diamonds is proportional to the number of distant metastases. The number (*N*) of patients and distant metastases (DM) in each group is indicated. Hazard ratios were estimated with a Cox proportional hazards multivariable model, adjusted for Grade, Ki-67, ERBB2 status, estrogen/progesterone receptor status, tumor size (pT), number of positive lymph nodes (pN), and age at surgery (as appropriate). The number of patients in each subgroup analysis corresponds to those reported in Supplementary Tables 1 and 2 for IHC and RT-qPCR, respectively. *P* value, Wald test *P* value. See also Supplementary Tables 1–3[14]. Source data are provided as a Source Data file.

At the genetic level, *EPN3* is situated in a region on the long arm of Ch. 17, roughly 10.8 Mbp apart from the *ERBB2* gene, which is also frequently amplified/overexpressed in BC[37,38]. The two alterations, however, seem to influence the tumor phenotype independently, as supported by finding that ~50% of the *EPN3*-amplified cases do not show *ERBB2* amplification. In addition, the major alteration of EPN3 is represented by overexpression without gene amplification, at a frequency (30–40%) that exceeds that of ERBB2 amplification (~18–20%)[38,39]. Consistently, EPN3 overexpression stratified poor prognosis BCs even in the ERBB2-negative subset. It is interesting, however, that in around one quarter of all *ERBB2*-amplified cases, there is concomitant amplification of *EPN3*. The co-occurrence of alterations becomes even more evident if one considers EPN3 overexpression (regardless of amplification) and ERBB2-positivity. In this case, the frequency climbs to almost two thirds of all ERBB2-positive cases. Thus, the cooperation between ERBB2 and EPN3 in BC biology appears worthy of future investigations, also in light of the possibility that EPN3 status might modify the response to anti-ERBB2 therapies[40]. Furthermore, EPN3 overexpression stratified poor outcome patients also in the BC node-negative category. Since recommendations for adjuvant therapy in this category of patients is not as straightforward as in the node-positive subgroup, the identification of EPN3 as an independent risk marker might aid in the development of more precise criteria for the eligibility for therapy in node-negative patients.

At the biological level, EPN3 contributes to mammary tumorigenesis likely through the induction of a partial EMT phenotype, characterized by the simultaneous presence of epithelial and mesenchymal markers[5,6,17,18,41–43]. Indeed, when compared with a potent inducer of EMT, TWIST, EPN3-overexpressing cells displayed a milder EMT phenotype, characterized by a less pronounced upregulation of the EMT transcription factors ZEB1 and TWIST, and not resulting in the transcriptional repression of ECAD, which was—however—regulated at the post-transcriptional level. The hypothesis that the EPN3-induced phenotype represents a partial EMT is also supported by the synergy of EPN3 and TGFβ towards a more advanced EMT state.

Interestingly, cells in a partial EMT state display the highest level of plasticity, i.e., the ability to switch between invasive/mesenchymal and proliferative/epithelial modes, when compared with cells that have undergone full EMT. Therefore, it has been proposed that cells harboring partial EMT are more prone to metastasize to distant organs[43]. This behavior correlates well with the clinical impact of EPN3 overexpression in the natural history of BC, which is exerted at the level of the metastatic phenotype rather than on local expansion and aggressiveness, as shown by the lack of correlation with loco-regional recurrence.

At the mechanistic level, one physiological function of EPN3, uncovered in our study, is to participate to ECAD internalization, in line with the general endocytic role attributed to epsins. However, there seems to be specificity in EPN3 molecular workings, as its perturbation did not affect the internalization of selected PM cargoes. In turn, this might be consistent with the restricted physiological pattern of EPN3 expression and its selective activation when the migratory phenotype is required[8], a condition in which loss of ECAD-mediated cell–cell contacts would be desirable. EPN3 overexpression resulted in an exaggeration of this physiological function, leading to increased ECAD internalization.

Data obtained with an inducible EPN3 overexpression system support the possibility that stimulation of ECAD endocytosis is indeed the first event in EPN3-driven EMT. This conclusion is in agreement with extant literature showing that: (i) at steady state, the slow, but continuous, endocytic uptake and recycling of ECAD (as herein seen, in unperturbed MCF10A cells, Fig. 3c) is involved in adherent junction remodeling[44,45], a process that has been shown to be critical in the establishment of epithelial cell polarity, epithelial cell division and tissue integrity[44,45], (ii) mechanisms that accelerate ECAD endocytosis and turnover at the PM are known to induce EMT and the invasive/metastatic phenotype[3], (iii) partial EMT phenotype is linked to the endocytic-dependent re-localization of epithelial markers, such as ECAD, and not to their direct transcriptional downregulation[17,41,42].

Our data do not allow to establish the mechanism of ECAD internalization regulated by EPN3. Indeed, although EPN3 binds to clathrin in MCF10A, this does not implicate that it drives ECAD endocytosis via clathrin-coated pits. Indeed, Epsins, despite their strong binding to clathrin and to the clathrin adaptor AP2, are known to regulate both clathrin-dependent and -independent mechanisms[22–24]. Furthermore, there is considerable functional redundancy among epsins[46,47]. While more work is needed to establish the ECAD internalization route controlled by EPN3, our data converge on the possibility that this function might be non-redundant at least vs. EPN1. This, in turn might have implications for the transforming potential of the various members of the family, as will be further discussed below.

Downstream of ECAD internalization, we dissected the molecular workings of the EPN3-triggered circuitry and showed that it is based on the activation of a TGFβ-based autocrine loop (Fig. 6e). The establishment of TGFβ-dependent auto-regulatory loops has been previously described as relevant to the induction of EMT[48,49], through the activation of large transcriptional programs that depend on a number of pathways, including β-catenin-, Wnt-, and RTK-based ones[3]. Consistent with this, we showed that, consequent to EPN3-induced ECAD endocytosis, the β-catenin/TCF4 pathway was directly involved in the

acquisition of the transcriptional circuitry necessary to sustain the partial EMT[25,26]. Among the distal events, the increased transcription of TGFβ ligands and receptors is of particular interest, as they could potentially generate an autocrine loop, sustaining and reinforcing the EPN3-triggered events, since TGFβ is in itself an inducer of ECAD endocytosis and of activation of EMT programs[28]. This possibility was formally proven by ablating the expression of TGFβRs, which reverted the EPN3-induced phenotypes. We acknowledge, however, that the contribution of other mechanisms is not excluded by our data. In particular, EPN3 overexpression might inhibit negative feedback regulators of TGFβ signaling (see, for instance, see ref. [50]). Furthermore, our data are also compatible with the existence of some EPN3-specific effects, independent of TGFβ signaling and of TCF4/β-catenin axis, which could act in parallel with the TGFβ-dependent ones.

Finally, we would like to comment on the oncogenic potential of the epsin family. While a potential role in regulating tumor angiogenesis has been proposed for EPN1 and EPN2[51,52], our results show that EPN1 overexpression does not phenocopy any of the investigated EPN3 effects, thereby pointing to a specific modality of transformation by EPN3, at least in BC. At the sequence level, EPN3 shows a high degree of similarity with the other family members. All three members possess binding motifs for clathrin, AP2, and ubiquitin. The central and the C-terminal portions of the protein are the most divergent ones. EPN3 displays a specific sequence, located after the ubiquitin-binding motifs, encoded by a specific exon (Exon 5) conserved between human, mouse, and rat, which is not present in EPN1-2. This region, therefore, is an obvious candidate for the identification of the molecular determinants of EPN3 specificity. In the future, it will be of interest to identify the minimal EPN3 region responsible for the effects, with the perspective of developing specific inhibitors. This, coupled to its restricted pattern of expression in adult tissues, makes EPN3 an attractive therapeutic target in BC. Furthermore, the interplay between EPN3- and TGFβ-based signaling in BC might be of clinical relevance, since several TGFβ signaling inhibitors are being developed[53]. EPN3 may, therefore, represent a predictive biomarker to identify which patients are most likely to benefit from targeted anti-TGFβ signaling therapies.

## Methods
**Description of the retrospective consecutive IEO BC cohort.** For the analyses reported in Fig. 10c, d and Supplementary Tables 1–3, we used a large retrospective consecutive cohort of 2453 breast cancer (BC) patients operated at the European Institute of Oncology (IEO), Milan, Italy, between 1997 and 2000 (the IEO BC 97-00 cohort). Detailed description of the selection criteria and the clinico-pathological characteristics of this cohort have been reported elsewhere[13]. Briefly, available clinical and pathological information included age, date at surgery, tumor characteristics (histological type, tumor size, nodal involvement, grade, perivascular infiltration, Ki-67 and ER/PgR expression), and treatment modality (type of surgery, adjuvant radiotherapy, endocrine therapy, chemotherapy). Patients of this cohort were followed up with physical examination every 6 months, annual mammography and breast ultrasound, blood tests every 6–12 months, and further evaluations only in the case of symptoms. When possible, the status of women not presenting at the institute for scheduled follow-up visits for more than one year was obtained by telephone contact. Cumulative incidence of Loco-regional (LR) and distant metastasis (CI-DM) were defined as the time from surgery to the appearance of a local or regional recurrence and distant metastases or death from BC as first event, respectively[54]. Second primary cancer or death from unknown causes or other causes were considered competing events. Considering first events, median follow-up for censored patients was 14.1 years (interquartile range [IQR] 12.1–15.7). One hundred and eighty-five (7.5%) patients were lost at 10 years of follow-up. Research on human samples was approved by the IEO Institutional Review Board.

**TMA construction and RNA extraction from the IEO BC cohort.** From the IEO BC 97-00 cohort of 2453 samples (see above), 2275 were appropriate for the construction of ad hoc tissue microarrays (TMAs): samples with massive

inflammatory infiltration, massive necrosis, minimal areas of infiltrating carcinoma were discarded. TMAs were prepared as previously described[55]. Briefly, representative tumor areas (three tissue cores of 0.6 mm in diameter) from each sample, previously identified on hematoxylin and eosin (H&E)-stained sections, were removed from the paraffin donor blocks and deposited on recipient blocks using a custom-built precision instrument (Tissue Arrayer - Beecher Instruments, Sun Prairie, WI 53590, USA).

For the PCR analysis, 2335 tissue blocks were appropriate for RNA extraction, and one tissue core of 1.5 mm diameter (in close proximity to the tissue core used for TMA construction whenever possible) or at least two tissue sections (10-µm thick) were taken from representative tumor areas with adequate tumor cellularity (>60%) selected by a pathologist for each tissue block. Samples with massive inflammatory infiltration, massive necrosis, minimal areas of infiltrating carcinoma were discarded. The total RNA was extracted from formalin-fixed paraffin-embedded (FFPE) tissues using the AllPrep DNA/RNA FFPE Kit automated on QIAcube, following the manufacturer's instructions (Qiagen, Hilden, Germany).

**Quantitative real-time PCR analysis of human BCs.** For *EPN3* mRNA analysis, 250 ng of the total RNA (RNA concentration measured using the NanoDrop® ND-1000 Spectrophotometer) were reverse transcribed with random primers using the SuperScript® VILO™ cDNA Synthesis Kit (Thermo Fisher Scientific), pre-amplified with the PreAMP Master Mix Kit (Thermo Scientific) for ten cycles, following the manufacturer's instructions, and diluted 1:25 prior to PCR analysis (5 µl were then used per PCR reaction, corresponding to 1 ng of cDNA). Quantitative PCR was performed with hydrolysis probes (Thermo Fisher Scientific) using the SsoAd-vanced Universal Probes Supermix (Bio-Rad Laboratories) in 10 µl of final volume in 384-well plates. PCR reaction was run in LightCycler (LC) 480 real-time PCR instruments (Roche) using the following thermal cycling conditions: 1 cycle at 95 °C for 30 s, 45 cycles at 95 °C for 5 s and 60 °C for 30 s.

TaqMan gene expression assays, with short amplicon sizes, were as follows: Hs00978957_m1 (EPN3), Hs02800695_m1 (HPRT1), Hs03929097_g1 (GAPDH), Hs99999908_m1 (GUSB), and Hs00427621_m1 (TBP).

We defined Cq=35 as our limit of detection. Therefore, Cq values beyond this limit were set to 35, and normalization was omitted. Each target was assayed in triplicate, and average Cq (AVG Cq) values were calculated either from triplicate values when the standard deviation was <0.4, or from the best duplicate values when the standard deviation was ≥0.4. Data (AVG Cq) were normalized using four reference genes (*HPRT1*, *GAPDH*, *GUSB*, and *TBP*) to account for variation in the expression of single-reference genes and in RNA integrity due to tissue fixation. The normalized Cq ($Cq_{normalized}$) of EPN3 target gene was calculated using the following formula:

$$Cq_{normalized} = AVG\ Cq - SF,$$

where SF is the difference between the AVG Cq value of reference genes for each patient and a constant reference value K; K represents the mean of the AVG Cq of the four reference genes calculated across all samples (K = 25.012586069). This normalization strategy allows the retention of information about the abundance of the original transcript, as measured by PCR (i.e., in Cq scale), which is conversely lost when using the more classical ΔCq method. Normalized data were then processed for statistical analysis. Based on the distribution of the reference genes, we applied the Tukey's interquartile rule for outliers[56] to identify poor-quality RT-qPCR data. In total, 16 samples (0.7%) were excluded from downstream analyses, either for lack of sufficient RNA or because of spurious RT-qPCR results (due to poor-quality mRNA). Therefore, 2319 patients in total (99.3%) were included in the statistical analysis. EPN3 mRNA status was stratified into low and high, based on the median of Cq normalized value, high EPN3 mRNA ≥28.18 or low EPN3 mRNA <28.18, respectively (Fig. 10c, d; Supplementary Tables 2 and 3).

**EPN3 expression analysis in the IEO BC cohort by IHC on TMA.** Three micrometer sections from the TMA blocks were cut, mounted on glass slides, and processed for IHC analysis using the in-house generated anti-hEPN3 mouse monoclonal antibody (Ab, clone VI31, epitope: aa residues 464–483, Homosapiens)[9] at a final concentration of 0.04 µg ml⁻¹. Following antigen retrieval in 1.0 mM EDTA pH 8.0, immunocomplexes were visualized by the EnVision™ + HRP mouse (DAB + ) kit, DAKO (Catalog number: K4007), and acquired with the Aperio ScanScope system (v12.2.2, Leica Biosystems). For scoring, a semi-quantitative approach, with scores ranging from 0 (negative staining) to 3 (intense staining) in 0.5 unit increments, was used. EPN3 expression, measured in parallel in non-neoplastic mammary gland areas of breast tissue samples (N = 32), was weak (IHC score ≤1.0, median score = 0.5) in >90% of the cases (29/32); tumor samples with IHC scores >1.0 were considered EPN3-high, and those with ≤1.0 were considered EPN3-low. Of 2275 cases arrayed on TMA, reliable IHC data for EPN3 expression was obtained in 1817 (79.9%) cases, mainly due to loss of tissue cores during staining and/or lack of neoplastic tissue in the tissue core (Fig. 10c, d; Supplementary Tables 1 and 3).

IHC analysis of ERBB2 expression was repeated ad hoc, in this cohort, for this study. Following antigen retrieval (1.0 mM EDTA pH 8.0, for 50 min at 95 °C), TMA sections were incubated with a rabbit polyclonal anti-ERBB2 Ab (dilution 1:1000, 0.6 µg ml⁻¹) from DAKO (Catalog number: A0485). Immunocomplexes were visualized by the EnVision™ + HRP Rabbit (DAB + ) kit, DAKO (Catalog

number: K4011), and acquired with the Aperio ScanScope system (v12.2.2, Leica Biosystems). ERBB2 expression was evaluated according to the HercepTest™ (DAKO) scoring system.

**FISH analysis of BC TMAs.** The FISH analysis (Fig. 1b) was performed on a cohort of BC patients available in the lab as TMAs[57,58] and derived from a clinical study (Trial registration ID: NCT00970983)[12]. The PathVysion HER-2 DNA Probe Kit (Abbott Molecular Inc.) was employed to detect ERBB2 amplification on TMA sections following the manufacturer's instructions. To analyze EPN3 amplification, the genomic clone CTD-2530L10, encompassing the hEPN3 gene, was selected from the CalTech human BAC library D (CTD) (Thermo Fisher Scientific). DNA probe was labeled with a fluorescent dye (Cy3-dUTP) (GE Healthcare Life Sciences) by nick translation[59]. Briefly, 1 µg of DNA probe was incubated for 2 h at 16 °C in a mix (final volume, 30 µl) containing 50 mM Tris-HCl, 5 mM $MgCl_2$ and 0.005% BSA, 4 µM dAGC (4 µM of dATP, dGTP, and dCTP each), 2.5 µM Cy3-dUTP, 10 mM ß-mercaptoethanol, 0.3 µl DNA polymerase (20,000 units, Thermo Fisher Scientific), 6 µl DNase (~0.017 units). The probe was ethanol-precipitated in the presence of 3 µg salmon sperm DNA and 10 µg human Cot-1 DNA (Thermo Fisher Scientific). After centrifugation, the dried pellet was resuspended in hybridization mix (50% formamide, 10% dextran sulfate, and 2xSSC). The mixture was then incubated with shaking for 10 min at RT. Once labeled, the DNA probe was hybridized on dried slides of the BC TMAs, following the protocol of the PathVysion HER-2 DNA Probe Kit (Abbott Molecular Inc.). Hybridized slides were analyzed using fluorescent microscopy.

EPN3 was considered amplified when the EPN3/CEP17 ratio was >2.5; ERBB2 was considered amplified when the ERBB2/CEP17 ratio was ≥2.0 according to Wolff et al.[60]. Reliable data on EPN3 and ERBB2 amplification were obtained on 219 samples (Fig. 1b). Cases in which results could not be obtained were due to loss of tissue cores during staining and/or lack of neoplastic tissue in the tissue core.

On the same cohort, we performed the initial comparative analysis of EPN3 amplification vs. overexpression (Fig. 1c, d). The procedures for EPN3 IHC are described above. In this case, we obtained reliable FISH and IHC results in 212 cases (Fig. 1d).

**Constructs and plasmids.** Full-length human EPN3 was isolated by RT-qPCR from MDA-MB-361 cells and sequence-verified, followed by subcloning, as appropriate, into the pBABE-Flag-puro retroviral vector (Figs. 2, 3, 4, 5, 7; Supplementary S3, S4, S5, S7, S8, S9), pLVX-puro lentiviral vector (Supplementary Figs. 2E and 6) or inducible pSLIK-neo lentiviral vector (gift from Iain Fraser, Addgene plasmid # 25735; http://n2t.net/addgene:25735; RRID:Addgene_25735, Fig. 7 and Supplementary Fig. 9C)[61]. Nucleotide sequences of the primers used for the cloning:

EPN3 Reverse: GAAGATCTACGACCTCCGCACTCCGG
EPN3 Forward: GGAATTCTCAGAGGAAGGGGTTGGTGCC.

Full-length human EPN1 was isolated by RT-qPCR from MDA-MB-361 cells and sequence verified, followed by subcloning, as appropriate, into the pBABE-Flag-puro retroviral vector.

pBABE-puro-mTwist was a gift from Bob Weinberg (Addgene plasmid # 1783; http://n2t.net/addgene:1783; RRID:Addgene_1783)[15].

The pSicoR lentiviral vectors (gift from Tyler Jacks, Addgene plasmid # 11579; http://n2t.net/addgene:11579; RRID:Addgene_11579)[62] were used for constitutive shRNA expression in DCIS.com cells (shEpn3#1, Fig. 9e, f; Supplementary Fig. 10B, C) and in BT474 (shEpn3#1 and 2, Supplementary Fig. 2C, D). Vectors were engineered to express shRNA specifically targeting Epn3 expression (shEpn3#1, shEpn3#2) or luciferase (shLuc) and mismatch (shMis) as controls. The following shRNA oligo sequences were used:

shEpn3#1 Forward:
TGCGAGAACCTCTACACCATTTCAAGAGAATGGTGTAGAGGTTCTCGCT
TTTTTC
shEpn3#1 Reverse:
TCGAGAAAAAAGCGAGAACCTCTACACCATTCTCTTGAAATGGTGTA
GAGGTTCTCGCA
shEpn3#2 Forward:
TGAGCTAGAAACTGAACGCCTTCAAGAGAGGCGTTCAGTTTCTAGC
TCTTTTTTC
shEpn3#2 Reverse:
TCGAGAAAAAAGAGCTAGAAACTGAACGCCTCTCTTGAAGGCGTTCA
GTTTCTAGCTCA
shOligoMis#1 Forward:
TGAGCGAACCGATACACTATTTCAAGAGAATAGTGTATCGGTTCGCT
CTTTTTTC
shOligoMis#1 Reverse:
TCGAGAAAAAAGAGCGAACCGATACACTATTCTCTTGAAATAGTGTA
TCGGTTCGCTCA
shOligoMis#2 Forward:
TGTGATAGGATCTGAACTCCTTCAAGAGAGGAGTTCAGATCCTATCA
CTTTTTTC
shOligoMis#2 Reverse:
TCGAGAAAAAAGTGATAGGATCTGAACTCCTCTCTTGAAGGAGTTCA
GATCCTATCACA.

**Cell culture.** All human breast cell lines were from the American Type Culture Collection (ATCC), with the exception of MCF10DCIS.com, which were kindly provided by Dr. John F Marshall (Barts Cancer Institute, London, UK)[63]. All human cell lines were authenticated at each batch freezing by STR profiling (StemElite ID System, Promega), and tested for mycoplasma by PCR[64] and biochemical assay (MycoAlert, Lonza).

MCF10A and MCF12A cell lines were cultured in a 1:1 mixture of DMEM and Ham's F12 medium (Gibco, Life Technologies), supplemented with 5% horse serum (Invitrogen), 2 mM L-glutamine, 20 ng ml$^{-1}$ human EGF (Invitrogen), ng ml$^{-1}$ cholera toxin, 10 µg ml$^{-1}$ insulin, and 500 ng ml$^{-1}$ hydrocortisone (Sigma). The HCC1569 cell line was cultured in the RPMI-1640 medium (Lonza), supplemented with 10% FBS and 2 mM L-glutamine. BT474, MCF7, SKBR3, MDA-MB-453, and MDA-MB-361 cells were cultured in the DMEM medium (Lonza), supplemented with 10% FBS and 2 mM L-glutamine (Euroclone). The HMEC cell line was cultured in MEGM medium, supplemented with Bullet kit (Lonza). MCF10DCIS.com were grown in the same culture medium of MCF10A and MCF12A, in the absence of cholera toxin. All cells were cultured at 37 °C in a humidified atmosphere containing 5% $CO_2$.

MCF10A and MCF12A cells infected with stably overexpressing retroviral constructs (pBABE-EPN3, -EPN1, -TWIST) or with pBABE empty vector, EV (referred as Ctr) were selected with 2 µg ml$^{-1}$ puromycin; HCC1569 cells infected with stably overexpressing lentiviral constructs, pLVX-EPN3 and pLVX-EV (referred as Ctr), were selected with 1.5 µg ml$^{-1}$ puromycin.

Unless otherwise indicated, cells at low/medium confluency (30–60%) were stimulated with different doses of TGFβ1 (0.75 or 5 ng ml$^{-1}$) for the indicated time points. Treatment with TGFβR kinase inhibitor LY2109761 (5 µM for 72 h), or with the same concentration of the vehicle DMSO, was performed on MCF10A cells at 40–50% confluency.

To generate doxycycline-inducible EPN3 expressing cells (Fig. 7), MCF10A-Ctr cells were infected with pSLIK-neo lentiviral construct expressing EPN3 (referred as EPN3$^{dox}$) or as pSLIK-neo empty vector (Ctr$^{dox}$). Selection was performed with 150 µg ml$^{-1}$ neomycin. To induce EPN3 expression, cells were treated with 200 ng ml$^{-1}$ doxycycline for 6 and 24 h.

**Reagents and antibodies.** Human recombinant TGFβ1 was from Peprotech; LY2109761 was purchased from Selleckbio. Duolink in situ Proximity Ligation Assay (PLA) was from O-link Bioscience (Duolink In Situ Detection Reagents Orange), and was performed according to the manufacturer's directions.

Antibodies used in the study are described in Supplementary Table 4 (primary antibodies) and Supplementary Table 5 (secondary antibodies).

**RNA interference (RNAi).** RNAi was performed with Lipofectamine RNAimax reagent from Invitrogen, according to the manufacturer's instructions.

For transient KD of EPN3 (Fig. 5f, g), cells were subjected to single reverse transfection with 10 nM RNAi oligos and analyzed 3 days after transfection. The following RNAi oligo (Dharmacon) was used: GUACAAGGCUCUAACAUUG. As a control, the following non-targeting siRNA oligo (Dharmacon) was used (D-001810-01-05): UGGUUUACAUGUCGACUAA.

For TCF4 KD (Figs. 4f, 6c, d; Supplementary Fig. 8D), cells were subjected to two cycles of reverse transfection at day 0 and day 4 with 20 nM of iBONI siRNA pool (Riboxx), with the following sequences:
AUAAUACAGAACCAACUCCCCC
UUCUUCCAAACUUUCCCGGCCCCC
UUAAAGUCUGCUGCCUACCCCC.

For transient KD of TGFβ receptors (Figs. 6c, d, 8f, g; Supplementary Fig. 8B, D), cells were subjected to double transfection (reverse) with 8 nM smart pool of RNAi oligos and analyzed 4–7 days after transfection. As negative control siRNA, we used "All Stars" from Qiagen or mock-silenced cells.

The following RNAi oligos from Dharmacon were used:
- TGFBR1 smart pool:
GAGAAGAACGUUCGUGGUU
UGCGAGAACUAUUGUGUUA
GACCACAGACAAAGUUAUA
CGAGAUAGGCCGUUUGUAU
- TGFBR2 smart pool:
CAACAACGGUGCAGUCAAG
GACGAGAACAUAACACUAG
GAAAUGACAUCUCGCUGUA
CCAAUAUCCUCGUGAAGAA

**Quantitative real-time PCR analysis.** The total RNA was extracted from cells (Figs. 2d, 5d, 5g, 6a, 6d, 7c; Supplementary Figs. 2A, 3B, 5F, 6C, E, 7B, 8A, B, D, 9A, C) using the RNeasy kit from Qiagen, according to the manufacturer's protocol. Single-stranded cDNA synthesis was performed using the SuperScriptII reverse transcriptase (Invitrogen) following the manufacturer's instructions. Briefly, 1 µg of the total RNA was mixed with 250 nM of random primers in RNase-free water and then incubated at 70 °C for 5 min. Following incubation, 10× reaction buffer, dNTPs mix (0.5 mM final concentration), and 1 µl of reverse transcriptase were added to the mix (20 µl final volume), and the reaction was incubated at 42 °C for

1 h. Finally, the reaction was inactivated by heating at 70 °C for 15 min. Taqman gene expression assays were as follows: Hs00978957_m1 (EPN3), Hs00170423_m1 (CDH1), Hs00169953_m1 (CDH2), Hs00185584_m1 (VIM), Hs00998133_m1 (TGFB1), Hs00234244_m1 (TGFB2), Hs00610320_m1 (TGFBR1), Hs00234253_m1 (TGFBR2), Hs00232783_m1 (ZEB1), Hs99999903_m1 (ACTB), Hs99999901_s1 (18 S), and Hs99999905_m1 (GAPDH).

Unless otherwise indicated in the figure legend, the data were normalized on the average expression of the housekeeping genes 18S-ACTB-GAPDH, and reported as relative fold change compared with Ctr/not treated cells, mean ± S.D.

The list of primers used in the study can be found in Supplementary Table 6.

**EPN3 transcripts and copy number analysis in breast cells**. The total RNA (Supplementary Fig. 2A) and genomic DNA (Supplementary Fig. 2B) were simultaneously extracted from fresh/frozen (FF) breast cell lines using the AllPrep DNA/RNA/miRNA Universal Kit (Qiagen, Hilden, Germany) automated on QIAcube, following the manufacturer's instructions (Qiagen, Hilden, Germany).

For EPN3 and ERBB2 mRNA analysis, 500 ng of the total RNA (RNA concentration measured using the NanoDrop® ND-1000 Spectrophotometer) were reverse transcribed with random primers using the SuperScript® VILO™ cDNA Synthesis Kit (Thermo Fisher Scientific), and 5 ng of cDNA/reaction was analyzed by PCR. Quantitative PCR was performed with hydrolysis probes (Thermo Fisher Scientific) using the SsoAdvanced Universal Probes Supermix (Bio-Rad Laboratories) in 10 µl final volume in 384-well plates. PCR reaction was run in LightCycler (LC) 480 real-time PCR instruments (Roche) using the following thermal cycling conditions: 1 cycle at 95 °C for 30 s, 45 cycles at 95 °C for 5 s, and 60 °C for 30 s. TaqMan gene expression assays were as follows: Hs00978957_m1 (human EPN3, RefSeq NM_017957.2, exon boundary 5–6, assay location 1328, amplicon length 62 bp) and Hs03929097_g1 (human GAPDH, RefSeq NM_001256799, exon boundary 8–8, assay location 1250, amplicon length 58 bp). Custom TaqMan gene expression assay to detect ERBB2 mRNA was designed using the Primer Express software (v3.0, Thermo Fisher Scientific):

Forward primer: GGATGTGCGGCTCGTACAC
Reverse primer: TAATTTTGACATGGTTGGGACTCTT
Probe-FAM: ACTTGGCCGCTCGG

Each target was assayed in triplicate, and average Cq values were calculated (the average was calculated from triplicate values when the standard deviation was <0.4, or from the best duplicate values when the standard deviation was ≥0.4). Data (average Cq) were normalized to the average Cq value of the endogenous reference gene (GAPDH), and to MCF10A as calibrator sample using the comparative Cq ($2^{-\Delta\Delta Cq}$) method (Supplementary Fig. 2A). RT-qPCR experiments were performed three times.

For EPN3 and ERBB2 gene copy number analysis, we used real-time quantitative PCR. Genomic DNA (1 µg) was digested with HaeIII prior to PCR analysis. Digested DNA (10 ng) was analyzed using the SSAdvanced Universal Probes Supermix (Bio-Rad Laboratories) in 10 µl final volume in 384-well plates. PCR was run in LightCycler (LC) 480 real-time PCR instruments (Roche) using the following thermal cycling conditions: 1 cycle at 95 °C for 30 s, and 45 cycles at 95 °C for 5 s and 60 °C for 30 s. TaqMan copy number assays were as follows: Hs03036374_cn (human EPN3, Chr.17:48613920, amplicon length 77 bp) and Hs02803918_cn (human ERBB2, NG_007503.1, Chr.17:37881203, amplicon length 70 bp). Custom TaqMan copy number assay to control for chromosome 17 (CEP17) was designed using the Primer Express software (v3.0, Thermo Fisher Scientific):

Forward primer: TTGCAGCACGTGGCACAT
Reverse primer: ACGGCAGCAAGAGAGGAAAG
Probe-FAM: CACTGCCTGAGCACC

Three independent PCR experiments were performed for each cell line, and each target was assayed in triplicate. Average Cq values were calculated (the average was calculated from triplicate values when the standard deviation was <0.4, or from the best duplicate values when the standard deviation was ≥0.4). Each replicate of either EPN3 or ERBB2 was normalized to centromeric probe CEP17 to obtain a ΔCq, and then an average ΔCq for each sample (from the three replicates) was calculated. All samples were then normalized to MCF10A used as calibrator sample (2n genome for chromosome 17) to determine ΔΔCq. Relative quantity (RQ) is $2^{-\Delta\Delta Cq}$, and copy number is 2 × RQ (Supplementary Fig. 2B).

The list of primers used in the study can be found in Supplementary Table 6.

**Cell lysis and immunoblot (IB)**. Cells were lysed in RIPA buffer (50 mM Tris-HCl, 150 mM NaCl, 1 mM EDTA, 1% Triton X-100, 1% sodium deoxycholate, 0.1% SDS), supplemented with a protease inhibitor cocktail (CALBIOCHEM) and phosphatase inhibitors (20 mM Na-pyrophosphate pH 7.5, 50 mM NaF, 10 mM Na₃VO₄ pH 7.5). Lysis was performed directly in the cell culture plates using a cell-scraper, and lysates were clarified by centrifugation at 16,000 g for 10 min at 4 °C. Protein concentration was measured by the Bradford assay (Biorad) following the manufacturer's instructions. Proteins were transferred to nitrocellulose filters. Filters were blocked for 1 h (or overnight) in 5% milk in TBS (50 mM Tris-HCl, pH 7.4, 150 mM NaCl) supplemented with 0.1% Tween (TBS-T). After blocking, filters were incubated with the primary Ab, diluted in TBS-T 5% milk, for 1 h at RT or overnight, followed by three washes of 5 min each in TBS-T. Filters were then incubated with the appropriate horseradish peroxidase-conjugated secondary Ab

(anti-mouse IgG HRP-linked 7076 or anti-rabbit IgG HRP-linked 7074, Cell Signaling) diluted 1:2000 in TBS-T for 30 min. After three washes in TBS-T (5 min each), the bound secondary Ab was revealed using the ECL method (Amersham) with Image Lab software (v3.0, Bio-Rad Laboratories).

For several IB experiments, samples were loaded on different gels (since many proteins have similar molecular weight), which were subsequently cut in the region of interest to allow detection with multiple Ab. Reblotting was also performed in some cases (see Source Data for details).

Quantitations of IBs in Fig. 2c and Supplementary Fig. 8C were obtained through the Gel Analysis Plug In of ImageJ software (v1.52, NIH). Each sample was normalized to its corresponding loading control and expressed as relative fold change compared with its control sample, as specified in legends to Fig. 2c and Supplementary Fig. 8C.

**Co-immunoprecipitation (Co-IP)**. For co-IP experiments (Fig. 4d), MCF10A-Ctr and -EPN3 cells were plated on 150-mm plates. After 48 h, cells were washed twice in PBS and EGF starved overnight. Cells (at a confluence between 80 and 95%) were then washed twice in PBS and subjected to cross-linking (formaldehyde 1% in ddH₂O) for 10 min at RT with shaking. Buffer was removed and a glycine buffer (glycine 0.125 M in ddH₂O) was added to cells for 5 min at RT with shaking. Cells were washed twice with cold PBS and lysed in RIPA buffer by scraping directly on plates (300 µl RIPA buffer/150-mm plate, see "Cell lysis and immunoblot" paragraph for detailed information). Cells were left 30 min on ice, followed by sonication (20 cycles at 4 °C: 30 s of sonication and 30 s of pause) and centrifugation at 16,000 g at 4 °C for 40 min (protocol adapted from ref. [65]). A pre-clearing step was performed by incubating the supernatants with Protein G for 1 h at 4 °C with rotation. Protein concentration was quantified in lysates with the Bradford Assay (Biorad) before performing the co-IP experiments. Co-IP was performed with the anti-EPN3 Rabbit polyclonal Ab, at a concentration of 5 ng mg⁻¹ of fresh lysate, in a volume of 400 µl. The lysate-Ab mixture was incubated for 2 h at 4 °C with rotation. Protein G was added and incubated for an additional 1 h. Samples were washed four times in RIPA buffer, and elution was performed in Laemmli buffer 2×.

**ECAD staining at the cell-to-cell junction**. MCF10A-Ctr, -EPN1, -EPN3, and TWIST cells were grown at confluency on coverslips for 48 h. Cell were fixed with 4% PFA and subjected to IF with anti-ECAD Ab (BD Bioscience) and DAPI staining (Fig. 4a; Supplementary Fig. 4A). Briefly, cells were permeabilized in PBS 0.1% Triton X-100 for 10 min at RT. To prevent nonspecific binding of the Ab, cells were incubated with PBS in the presence of 5% BSA for 30 min and then with primary Ab diluted in PBS (+ 0.2% BSA) for 1 h at RT. Coverslips were washed three times with PBS and incubated for 30 min at RT with secondary Ab Alexa-488-conjugated (green, donkey anti-mouse IgG, A-21202, Thermo Fischer, dilution 1:400). After three washes in PBS, coverslips were mounted in a 90% glycerol solution containing diazabicyclo-(2.2.2)octane antifade (Sigma). Images were obtained using a Leica TCS SP2 or TCS SP2 AOBS confocal microscope equipped with a 63× oil objective and processed using ImageJ software (v1.52, NIH).

**ECAD and EPN3 co-localization**. Co-localization between ECAD and EPN3 (Fig. 5c) was performed using an anti-ECAD Ab (HECD-1) from Abcam, recognizing the extracellular domain of ECAD[66]. Briefly, MCF10A-Ctr or -EPN3 cells were plated on coverslips at ~30% confluence. Cells were starved overnight in culture medium containing 5% horse serum in absence of EGF. After starvation, coverslips were washed with PBS and incubated with the anti-ECAD Ab (2 µg ml⁻¹) for 1 h at 4 °C. The time zero sample was immediately fixed with 4% PFA. The other samples were incubated at 37 °C for 90 min with culture medium containing 5% horse serum and TGFβ (5 ng ml⁻¹). After incubation cells were fixed with 4% PFA, followed by IF (as described in the Paragraph: "ECAD staining at the cell-to-cell junction"). Cells were stained with anti-EPN3 rabbit polyclonal Ab (50 ng ml⁻¹). EPN3 and ECAD signals were then revealed with secondary Ab Cy3-conjugated (red, donkey anti-rabbit IgG 715-165-152, Jackson ImmunoResearch, dilution 1:400) or Alexa-488-conjugated (green, donkey anti-mouse IgG 715-165-150, Jackson ImmunoResearch, dilution 1:400), respectively. Images were obtained using a Leica TCS SP2 or TCS SP2 AOBS confocal microscope equipped with a 63× oil objective and processed using ImageJ software (v1.52, NIH).

**IF analysis of active β-catenin**. MCF10A-Ctr and -EPN3 cells (150,000 cells) were plated on coverslips into six-well plates. After 48 h, cells were fixed/permeabilized in methanol for 10 min at −20 °C, washed with PBS, and blocked with 2% BSA to prevent nonspecific binding. Cells were stained with anti-active β-catenin (Millipore 8E7) diluted in PBS 0.2% BSA for 1 h. After three washes with PBS, cells were incubated for 1 h with secondary Ab (donkey anti-mouse Alexa-488-conjugated, A-21202, or donkey anti-mouse Alexa-647-conjugated, A-31571, Thermo Fisher, dilution 1:400). Cells were then stained with DAPI (Sigma) for 5 min. Finally, after three washes with PBS, coverslips were mounted on a 90% glycerol solution (Sigma) and examined under a confocal microscope (AOBS SP2, SP5 or SP8, Leica). Images were processed with ImageJ software (v1.52, NIH). For each image (.tiff), a DAPI mask was used to identify nuclei and to measure the mean fluorescence intensity of active β-catenin in the nucleus for each cell. Finally, active

β-catenin staining in nuclei was reported as relative fold change compared with Ctr cells (Figs. 4e and 7f, g).

**Analysis of surface ECAD signal intensity by FACS**. MCF10A-Ctr, -Mock, or -EPN3-KD cells ($8 \times 10^5$), or MCF10A-EPN3 cells ($1.2 \times 10^5$) were plated on 35-mm dishes. After 36 h, cells (at 60–80% confluency), were washed twice with PBS and incubated for 3 h with medium without EGF. Cells were stained with anti-ECAD Ab (Abcam, $2 \, \mu g \, ml^{-1}$) for 1 h at 4 °C, followed by incubation with secondary Ab (donkey anti-mouse IgG Alexa-488-conjugated, $5 \, \mu g \, ml^{-1}$, A-21202, Thermo Fisher) for 30 min at 4 °C. Cells were then washed twice with PBS and detached from the plates with trypsin 0.25% (15–20 min at 37 °C). Cells were recovered in medium in a 15 ml Falcon, washed once with PBS, resuspended in 500 μl PBS, and fixed by adding 500 μl of formaldehyde 2% for 15 min on ice. Subsequently, cells were centrifuged at 335 g for 5 min and resuspended in PBS with EDTA 2 mM before analysis with the FACS Celesta (BD). Samples stained only with secondary Ab were used as a negative control. Samples stained for ECAD and fixed or not with formaldehyde 2% were compared to evaluate the effect of fixation: no significant difference was observed both in fluorescence intensity and in number of cells positively stained. Analysis was performed using the FlowJo software (v10.4.2, LLC). Briefly, cell doublets and clumps were excluded through the side scatter area (SSC) over the height peak parameter. Cells with correct morphology were then selected through the forward scatter area (FSC) over SSC area parameters. Finally, ECAD mean fluorescence intensity of the population was used to compare the different samples.

**ECAD internalization assay by FACS**. FACS analyses were used to follow ECAD internalization (Figs. 4c, 5b, 5f, 7e). Cells were subjected to the same protocol described in "Analysis of PM-ECAD signal intensity by FACS". In this case, after the incubation with anti-ECAD and secondary Ab, cells were washed twice with PBS and re-incubated at 37 °C with complete medium ± TGFβ $5 \, ng \, ml^{-1}$. At the indicated time points, cells were washed once with cold PBS and treated with acid wash (AW) solution (glycine 0.1 M in ddH₂O, pH = 2.2) on ice for three times, 45 sec/each. Cells were then washed twice with PBS, and detached from the plates with trypsin 0.25% (15–20 min at 37 °C). Subsequently, cells were fixed and processed as described in "Analysis of PM-ECAD signal intensity by FACS". To evaluate the amount of internalized ECAD, the mean fluorescent intensity of internalized ECAD for each time point was divided by the total ECAD fluorescent intensity. The time 0 min was set as background, and its value subtracted from the other time points.

**ECAD internalization assays by IF**. IF was used to follow ECAD internalization (Figs. 5a, 7d). Cells were subjected to EGF-starvation for 3 h, then stained with anti-ECAD Ab (Abcam, $2 \, \mu g \, ml^{-1}$) for 1 h at 4 °C. Then, cells were washed twice with PBS and re-incubated at 37 °C with complete medium ± TGFβ $5 \, ng \, ml^{-1}$. At the indicated time points, cells were washed once with cold PBS and treated with acid wash (AW) solution (glycine 0.1 M in ddH₂O, pH = 2.2) on ice for three times, 45 s/each. Cells were then washed twice with PBS, fixed for 10 min with PFA 4% at RT, incubation with secondary Ab (donkey anti-mouse IgG Alexa-488-conjugated, $5 \, \mu g \, ml^{-1}$, A-21202, Thermo Fisher) for 30 min, by DAPI staining and confocal microscopy analysis (AOBS SP2, SP5 or SP8, Leica). Images were processed with ImageJ software (v1.52, NIH). For each image (.tiff), different fields of view for each image were isolated and quantified by Integrated Density (IntDen) to evaluate signal intensity for the different samples. The number of cells for each field of view was counted, and the IntDen was divided by the number of cells per field of view. Finally, ECAD intensity was reported as relative fold change compared with the indicated sample (Figs. 5a, 7d).

**$^{125}$I-EGF, $^{125}$I-TGFβ internalization and recycling assays**. MCF10A cells were plated in 24-well plates (50,000/well) in triplicate for each time point, plus one well to measure nonspecific binding. The day after, cells were starved for at least 4 h in MCF10A standard medium without EGF (supplemented with 20 mM Hepes and 0.1% BSA) and then incubated at 37 °C in the presence of $1.5 \, ng \, ml^{-1}$ $^{125}$I-EGF (Supplementary Fig. 5C, D) or $0.75 \, ng \, ml^{-1}$ of $^{125}$I-TGFβ (Supplementary Fig. 5A, B). At different time points (2, 4, 6 min), cells were placed on ice, washed twice in cold PBS, and then treated for 5 min at 4 °C in 300 μl of acid wash solution (0.2 M acetic acid, 0.5 M NaCl, pH 2.5). The radioactivity recovered in the acid wash represents the amount of ligand bound to the receptor on the cell surface. Cells were then lysed with 300 μl of 1 N NaOH, which represents the amount of internalized ligand. Nonspecific binding was measured at each time point in the presence of an excess of non-radioactive EGF (300 times) or TGFβ (500 times). After being corrected for nonspecific binding, the data were plotted as the ratio between internalized and surface-bound radioactivity over time. Recycling of $^{125}$I-TGFβ was assessed by starving MCF10A cells for at least 2 h in standard medium without EGF (supplemented with 0.1% BSA, 20 mM Hepes), and incubating cells with $0.75 \, ng \, ml^{-1}$ of $^{125}$I-TGFβ at 37 °C. Cells were then chased at 37 °C in a medium containing $375 \, ng \, ml^{-1}$ cold TGFβ for the indicated times to allow recycling and then processed following the protocol described for $^{125}$I-EGF recycling assay[67].

**ELISA assay**. ELISA assay to measure TGFβ1 concentrations in MCF10A cell culture medium (Fig. 6b; Supplementary Fig. 8A) was performed according to the manufacturer's instructions (Human TGFβ1 Quantikine ELISA Kit, R&D). Briefly, cells were plated in 150-mm dishes with 20 ml of complete medium, at least two dishes per each condition. When cells reached confluency, after 2 days from plating, they were serum starved for 24 h in 10 ml of complete medium without serum. The following day, cell culture media were concentrated, using Amicon Ultra 15 10 KDa cutoff tubes (Merck), to ~750 μl. Supernatants were then activated to detect latent TGFβ1 in the medium with 1 N HCl for 10 min, neutralized with 1.2 N NaOH/0.5 M HEPES, and immediately spotted onto microplates coated with monoclonal Ab anti-TGFβ1 for 2 h at RT. Wells were washed three times with the manufacturer's wash buffer and then treated with anti-TGFβ1 conjugated with horseradish peroxidase for 2 h at RT. After three more washes, substrate solution (hydrogen peroxide and stabilized chromogen tetramethylbenzidine) was added for 30 min at RT. Finally, stop solution (hydrochloric acid) was added, and plate absorbance was immediately read at Glomax (Promega) at the specific wavelength of 450 nm and at the nonspecific wavelength of 750 nm. Cell culture supernatants of each dish were assayed in triplicate, together with duplicates of known TGFβ concentrations (standards), which were used to create a standard curve and to calculate the equation of the curve and hence derive the concentration of TGFβ in cell culture supernatants. The concentration of TGFβ was divided by the number of cells on the day of the experiment and normalized for 10 million cells. Data were reported as pg ml⁻¹ 10⁻⁶ cells.

**CD44–CD24 flow-cytometry analysis**. Cells were stained with the anti-CD44 and anti-CD24 Ab diluted 1:5 in 1% BSA (100 μl/500,000 cells) and then fixed in 1% formaldehyde and analyzed by BD FACSCantoII (Fig. 8a, f; Supplementary Fig. 10A). BD FACSAria was used to sort live cells into CD44$^{high}$/CD24$^{low}$ and CD44$^{low}$/CD24$^{high}$ populations (Fig. 8d, e). Nonstained cells were used as a negative control to establish the background fluorescence. Gating on FSC compared with SSC was initially applied to isolate viable single-cell populations; then single-stained cells were used to set the gates for each dye used; finally, additional gating was used to select for CD44$^{high}$/CD24$^{low}$ population in Fig. 8a, f. The CD44$^{high}$/CD24$^{low}$ population is also represented as histograms for CD24 and CD44 in Supplementary Fig. 10A. Flow-cytometric analysis was performed using BD FACSDiva software (v8.0.1.1, BD).

For IB analysis on the sorted populations (Fig. 8e), CD44$^{high}$/CD24$^{low}$ and CD44$^{low}$/CD24$^{high}$ populations were re-plated after sorting, and lysed after 48 h. FACS analysis with CD44/CD24 markers was performed immediately after sorting to control for the purity of the two populations.

**Mammosphere assay**. Unsorted or sorted single-cell suspensions (10,000 cells ml⁻¹) were plated on ultralow attachment plates coated with Poly-HEMA, embedded in 1% methylcellulose in complete stem cell medium[35]: MEBM (Lonza) supplemented with 1% glutamine, 1% penicillin–streptomycin antibiotics, insulin $5 \, \mu g \, ml^{-1}$, hydrocortisone $0.5 \, \mu g \, ml^{-1}$, heparin $1 \, U \, ml^{-1}$, growth factors and supplement (freshly added): EGF $20 \, ng \, ml^{-1}$ (Peprotec), FGF $20 \, ng \, ml^{-1}$ (Peprotec) and B27 supplement (Gibco) (Fig. 8c, d). After 10 days, images of mammospheres were acquired through confocal microscopy (Leica TCS SP5 AOBS microscope system with Las-X software, v3.5 Leica Biosystems, equipped with Leica HyD and PMT detectors); spheres with diameter >50 μm were counted, and data were reported as sphere-forming efficiency (SFE: number of spheres/number of plated cells × 100).

**Matrigel MCF10A morphogenetic assay**. MCF10A cells were suspended in their culture medium containing $5 \, ng \, ml^{-1}$ EGF (instead of $20 \, ng \, ml^{-1}$ of standard medium) and 2% of growth factor reduced Matrigel (Corning). Cells (1000 cells/well) were plated on eight-well or four-well glass slides (chamber slide system, Lab-Tek II) on a thin layer of Matrigel (~60 μl, previously allowed to solidify on the bottom of the plate). Cells were incubated at 37 °C, 5% CO₂ for 14/21 days. Medium was changed every 3 days (Fig. 2f, g; Supplementary Fig. 3C, D). Images were acquired using a phase-contrast microscope at an initial time point of the morphogenetic program, day 5 (Supplementary Fig. 3C), and at completion of the morphogenetic program (days 14–21, depending on the experiment as described below, Fig. 2f, g; Supplementary Fig. 3D), and processed with ImageJ software (v1.52, NIH). For images acquired at day 5, *acini* were counted and measured for their diameter to discriminate organoids larger or smaller than 50 μm (Supplementary Fig. S3D). As the completion of the morphogenetic program was variable among experiments, samples were either stopped at 14 days or at 21 days in individual experiments. Images were acquired, and *acini* larger than 50 μm were counted. Circularity was measured by ImageJ software (v1.52, NIH) to quantify regular and irregular organoids (circularity larger or smaller than 0.8, respectively) in each experiment. Results are expressed as mean ± S.D. of independent experiments (see Fig. 2f, bottom).

For IF staining of the *acini* with Laminin V and Giantin (Fig. 2g), slides were fixed directly in eight-well chamber slides with 4% paraformaldehyde for 20 min at RT and permeabilized with 0.5% Triton X-100 in PBS for 10 min at 4 °C. For IF staining of the *acini* with ECAD and active β-catenin (Fig. 7g), slides were fixed directly in the four-well chamber slides with methanol for 20 min at −20 °C. Slides were rinsed three times with PBS/Glycine (100 mM), 10 min/wash at RT. Blocking

was in 10% goat or donkey serum in PBS (with 0.1% BSA, 0.2% Triton X-100, 0.05% Tween-20). Primary Ab anti-epiligrin (Laminin 5, Millipore), anti-giantin (Babco), anti-ECAD (BD Bioscience), and anti-active β-catenin (Millipore) were diluted in blocking solution and incubated overnight at 4 °C. Secondary Abs (donkey anti-mouse or anti-rabbit, Alexa-488- or Alexa-647-conjugated, Thermo Fisher; donkey anti-mouse or anti-rabbit Cy3, Jackson ImmunoResearch) were diluted 1:200 in blocking solution and incubated for 40 min at RT. Nuclei were counterstained with DAPI. Images were acquired on glass-wells with a Leica TCS SP2 or TCS SP2 AOBS confocal microscope with Las-X software (v3.5, Leica Biosystems) and processed using ImageJ software (v1.52, NIH). Active β-catenin staining in nuclei was quantified as described in Paragraph "IF analysis of active β-catenin", and is reported as relative fold change compared with EV (Fig. 7g).

**Soft-agar assay**. The bottom agar layer was prepared from a 5% agarose stock solution (GIBCO) that was dissolved in the microwave and mixed with culture medium at a final concentration of 1%. The solution was dispensed (4 ml/well) in six-well plates and allowed to harden at 4 °C for 30 min. Cells were trypsinized and counted, and a cellular suspension of 25,000 cells ml$^{-1}$ was prepared. Then, 1 ml of the cellular suspension was added to 3 ml of medium containing 0.4% of agar, and the total 4 ml of cells and agar were plated in six-well plates and incubated for 3 weeks (Supplementary Fig. 2C). Colonies were counted utilizing a phase-contrast microscope.

**Invasion assay**. Growth Factor Reduced Matrigel invasion chambers placed on 24-well plates from BD were used (Figs. 2e and 5e). Cells were trypsinized, resuspended in culture medium, and counted. In total, 40,000 cells were resuspended in 500 μl of culture medium and added to the top chamber of the transwell, and incubated at 37 °C for 24 h. In the case of TGFβ stimulation, culture medium supplemented with 0.75 ng ml$^{-1}$ TGFβ was added to the lower chamber. After 24 h of incubation, transwells were washed with PBS and fixed with PFA 4% for 20 min. Transwell filters were stained with crystal violet for 20 min, and then extensive washes with MilliQ H$_2$O were performed. Stained membranes were used to take photographs of the cells and for quantification.

**In vivo xenografts**. Cells were trypsinized and resuspended in medium, followed by centrifugation at 300×g for 5 min and resuspension in 10 ml of PBS. Following cell count, the desired number of cells was centrifuged again in at 300×g for 5 min and resuspended in 40 μl of a 1:1 Matrigel-PBS solution, and immediately injected into the inguinal mammary fat pad of NOD/SCID IL2R gamma-chain null (NSG) female mice (6–8-weeks old). BT474 cells, either silenced for EPN3 (KD, sh#1) or mock-silenced (shLUC), were injected into opposite inguinal mammary fat pads of a female NSG mice (300,000 cells/injection, $N = 8$, number of injected mice). Tumors were grown for 13 weeks before being explanted (Supplementary Fig. 2D). EPN3-overexpressing HCC1569 (EPN3) or control vector-infected (Ctr) cells were injected into opposite inguinal mammary fat pads of a female NSG mice (1,000,000 cells/injection, $N = 8$, number of injected mice). Tumors were grown for 43 days before being explanted (Supplementary Fig. 2E).

MCF10DCIS.com cells, either silenced for EPN3 (DCIS-shEPN3) or mock-silenced (DCIS-Ctr), were injected into inguinal mammary fat pad of NSG female mice (100,000 cells/injection; number of injected mice, $N = 19$ for DCIS-Ctr and $N = 21$ for DCIS-shEPN3). Tumors were grown for 2, 3, 4, or 5 weeks, as indicated in legends (Fig. 9b–f; Supplementary Fig. 10B, C).

In each case, all injections in NSG mice were performed by resuspending cells in 20 μl of 3:1 PBS-Matrigel mix and injecting them into the inguinal mammary fat pad of NSG female mice, after intraperitoneal anaesthetization with 150 mg kg$^{-1}$ tribromoethanol (Avertin). Mice were monitored by hand-palpation for tumor development. Tumor growth was measured with a Vernier caliper by applying the standard formula: tumor volume = (length × width$^2$)/2. Mice were sacrificed when tumors reached a dimension of 0.5–1 cm$^3$. Tumors were explanted and processed for cell extraction or inclusion in FFPE. $P$ values were calculated using the two-tailed Student's $t$ test.

**IHC analysis on MCF10DCIS.com xenografts**. Three-μm sections were prepared from formalin-fixed paraffin-embedded MCF10DCIS.com xenograft and dried at 37 °C overnight. The sections were processed with Bond-RX fully Automated stainer system (Leica Biosystems) according to the following protocol. First, tissues were deparaffinized and pre-treated with the Epitope Retrieval Solution 1 (pH 6) at 100 °C for 20 min for ECAD staining, 40 min for hEPN3, or Epitope Retrieval Solution 2 (pH 9) at 100 °C for 20 min for NCAD. After washing steps, peroxidase blocking was performed for 10 min using the Bond Polymer Refine Detection Kit (#DC9800, Leica Biosystems). Tissues were incubated for 30 min with mouse monoclonal Ab anti-ECAD (BD, #610182, 1:1500), mouse monoclonal anti-hEPN3 (clone VI31 in-house, 1:30000) or mouse monoclonal anti-NCAD (Dako, #M3613, 1:50) diluted in Bond Primary Ab Diluent (#AR9352). Subsequently, tissues were incubated with post primary and polymer for a total of 16 min, developed with DAB-chromogen for 10 min and counterstained with hematoxilin for 5 min.

For α-SMA and p63 Ab, we performed a double sequential automated staining; tissues were pre-treated with the Epitope Retrieval Solution 1 (pH 6) at 100 °C for 20 min, incubated with rabbit monoclonal anti p63 (Abcam, ab124762, 1:8000),

and subsequently with mouse monoclonal anti α-SMA Ab (Dako, # 80851, 1:200). Tissues were then incubated with post primary and polymers, respectively Polymer Refine Red Detection Kit (#DS9390) for p63 or Bond Polymer Refine Detection Kit (#DC9800) for α-SMA. Finally the sections were counterstained with hematoxilin for 5 min and acquired with the Aperio ScanScope system (v12.2.2, Leica Biosystems).

**Generation of inducible EPN3-KI mice**. An inducible EPN3 knock-in mouse model (EPN3-KI) was generated by the OzGene company (Bentley, Australia). The KI strategy is based on the targeted insertion of the human EPN3 cDNA (NM_017957.3) into the *ROSA26 locus*. Expression of the transgene is under the human Ubiquitin C promoter (UbiC), which is known to produce high-level expression in transgenic mice[68]. A loxP-flanked STOP cassette, which contains signals designed to prevent transcription of the gene of interest, is placed between the promoter and the coding sequence. The STOP cassette can be removed using Cre recombinase leading to the Inducible expression of the transgene. The EPN3-KI mice were back-crossed into FVB background to obtain FVB EPN3-KI mice for experiments.

**Mouse mammary bilayered organoid culture**. To obtain organoids with double acini layer, inguinal and thoracic mammary glands from EPN3-KI mice of 6–12 weeks of age were used. Bilayered organoids were prepared essentially as described in ref. [69], with modifications. Briefly, mammary glands were minced with scissors and partially digested on a rotating wheel (1×g) for 1 h at 37 °C (5% CO$_2$), in the following digestion medium: 1:1 mixture of DMEM and Ham's F12 medium (DMEM/F12, Gibco, Life Technologies), 1% penicillin–streptomycin antibiotics, trypsin 2.5% and collagenase type-1a 1 mg ml$^{-1}$ (Sigma-Aldrich, Merck Millipore). Then, mammary glands were centrifuged for 5 min at 350 ×g and a three-layers suspension was obtained. The liquid interface was discarded, while the fat layer (on the top) was transferred into a new tube, diluted with PBS and centrifuged for 10 min at 350×g. The pellet was resuspended in DMEM/F12 and added to the pellet of the first centrifugation. Samples were centrifuged again, and the pellet was treated with ACK lysis buffer (Gibco, Life Technologies) for 1 min to lyse blood cells, followed by dilution with PBS and centrifugation for 5 min at 350×g. The pellet was resuspendend in MEGM medium supplemented with Bullet kit (Lonza), and incubated for 30 min into a cell culture dish to allow the attachment of fibroblasts. Then, supernatant containing non-attached cells was passed through a 40-μm diameter filter, and the organoids trapped in the filter were resuspendend in MEGM medium and transferred into a nonadherent dish.

Organoids were treated or not with 100 μg ml$^{-1}$ of CRE in Optimem medium (Gibco, Life Technologies) for 2 h at 37 °C (5% CO$_2$). Then, they were centrifuged (350×g for 5 min), resuspended in MEGM medium and left overnight into a nonadherent dish. The CRE treatment was repeated the following day. After 5 days of recovery in nonadherent conditions in MEGM medium, organoids were then collected from the supernatant, counted, and resuspended in MEGM medium containing 2% of growth factor reduced Matrigel (Corning): 500–1000 organoids/well were plated in eight-well glass slides (chamber slide system, Lab-Tek II) on a thin layer of Matrigel (~60 μl, previously allowed to solidify on the bottom of the plate). Organoids were incubated at 37 °C, in 5% CO2 for 10–14 days. Medium was changed every 3 days (Fig. 3a, b; Supplementary Fig. 3E). Images were acquired using a phase-contrast microscope. Quantification was obtained by counting the number of invasive *vs.* non invasive organoids per field of view.

For IF staining of the organoids (Fig. 3a, b; Supplementary Fig. 3E), slides were fixed directly in the eight-well chamber slides with 4% paraformaldehyde for 20 min at RT. Permeabilization was performed with 0.2% Triton X-100 in PBS for 30 min at 4 °C and 30 min at RT. Slides were rinsed three times with PBS (10 min/wash at RT). Blocking was in 1% bovine albumin serum in PBS (with 0.1% BSA, 0.2% Triton X-100, 0.05% Tween-20). Primary Abs, anti-keratin-5, anti-keratin-8, anti-keratin-14, anti-ECAD, anti-NCAD, and anti-EPN3, were diluted in blocking solution and incubated overnight at 4 °C. Secondary Abs (donkey anti-mouse or anti-rabbit, Alexa-488- or Alexa-647-conjugated, Thermo Fisher; donkey anti-mouse or anti-rabbit Cy3, Jackson ImmunoResearch) were diluted 1:200 in blocking solution and incubated for 2.5 h at RT. Nuclei were counterstained with DAPI for 30 min. Images were acquired on glass-wells with Leica TCS SP2, TCS SP2 AOBS or TCS SP8 confocal microscope with Las-X software (v3.5, Leica Biosystems) and processed using ImageJ software (v1.52, NIH).

**Mouse study approval**. All mice have been maintained in a controlled environment, at 18–23 °C, 40–60% humidity and with 12-h dark/12-h light cycles, in a certified animal facility under the control of the institutional organism for animal welfare and ethical approach to animals in experimental procedures (Cogentech OPBA). All animal studies were conducted with the approval of Italian Minister of Health (27/2015-PR) and were performed in accordance with the Italian law (D.lgs. 26/2014), which enforces Dir. 2010/63/EU (Directive 2010/63/EU of the European Parliament and of the Council of 22 September 2010 on the protection of animals used for scientific purposes).

**Statistical analysis**. A logistic regression analysis was used to correlate EPN3 status with clinico-pathological parameters. The endpoints evaluated were CI-LR and CI-DM. The CI-LR and CI-DM functions were estimated according to methods described by Kalbfleisch and Prentice[70], taking into account the competing causes of recurrence. The hazard ratios (HR) comparison of EPN3-high *vs.* low tumors were estimated with a Cox proportional hazards multivariable model adjusted for Grade (G1-G2 and G3), Ki-67 (Ki-67 < 14% and Ki-67 ≥ 14%), ERBB2 status (positive and negative), estrogen/progesterone receptor status [not expressed (Both 0) and expressed (ER > 0 or PgR>0)], tumor size (pT1a/b, pT1c, pT2, pT3/pT4), number of positive lymph nodes (pN0, pN1-2-3 and pNX) and age at surgery (<50 and ≥50) (as appropriate in each subgroup analysis). All observations were truncated at 10 years from surgery. Analyses were carried out with the SAS software (SAS Institute, Cary, NC) and the R software (v3.5, cran.r-project. org/) with the "cmprsk" package developed by Gray (biowww.dfci.harvard.edu/ _gray). All reported *P* values are two-sided and *P* values <0.05 are considered as significant.

Statistical analyses for RT-qPCR experiments, IB and IF quantifications, FACS analyses, ELISA assays, mammosphere assays, radioactive internalization experiments, Matrigel morphogenetic assays, soft agar, and invasion assays were performed with Student's *t* test two-tailed, on Excel Office 2019 software (v17.0, Microsoft), with Each Pair Student's *t* test two-tailed, on JMP software (v14, SAS Institute), or with two-sided Fisher's exact *t* test, on Graph Pad Prism 8, as indicated in each figure legend. *P* values are shown as follows: *, <0.05; **, <0.01; ***, <0.001; ns, not significant. Quantitative data are presented as bars showing the mean of independent biological replicates ± S.D. or as box plots. Each box plot (in Figs. 3a, 4e, 5a, 7f, 7g, 9d; Supplementary Fig. 2D, E) is defined by 25 and 75 percentiles, showing median, whiskers representing 10 and 90 percentiles and outliers if present.

**Reproducibility**. Experiments such as IB, IF, IHC, phase-contrast microscopy images of cells (in Figs. 1c, 2a, b, g, 3b, 4a, f, 5c, 6c, 7a, b, d, 8e, g, 9a, b, c, e, 10a; Supplementary Figs. 1A, 1B, 3A, 3D, 3E, 4A, 6A, 6B, 6D, 7A, 7C, 9B, 10C) are representative of at least two independent experiments with similar results, unless otherwise indicated.

**Reporting summary**. Further information on research design is available in the Nature Research Reporting Summary linked to this article.

## Data availability

All data are available in the paper, in the Supplementary Information, or in Source Data file. The source data underlying Figs. 1–10, Supplementary Figs 1–10 and Supplementary Tables 1–3 are provided as a Source Data file. Source data are provided with this paper.

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

## Acknowledgements

We thank Rosalind Gunby for critically reading the manuscript; Chiara Luise, Giovanna Jodice, and Barbara Giulini for technical assistance; Marilena Bicchieri for help with the mammosphere assay; Alberto Gobbi, Manuela Capillo, and the Mouse facility (Cogentech Società Benefit Srl, Milan) for mice handling; Tom Kirchhausen for reagents and suggestions. The results shown in Fig. 1b are in part based upon the data generated by the TCGA Research Network: http://cancergenome.nih.gov/. This work was supported by grants from the Associazione Italiana per la Ricerca sul Cancro (AIRC - IG 18988, IG 23060 to P.P.D.F. and MCO 10.000; IG 18621, IG 22821 and MultiUnit −5 per Mille-22759 to GS; IG 11904, IG 15538, and MCO 10.000 to S.P.); the Italian Ministry of University and Scientific Research (MIUR) to P.P.D.F. (Prot. 2015XS92CC); the Italian Ministry of Health to D.T. (RF-2013-02358446); the Worldwide Cancer Research (16-1245) to S.S.; the European Research Council (Advanced-ERC − 268836) to G.S.; the Cariplo Foundation (2008.2448 to M.V.; 2011-0596 to A.D.). G.G., I.S.L., and C.T. were supported by an AIRC fellowship. This work was also partially supported by the Italian Ministry of Health with Ricerca Corrente and 5 × 1000 funds.

## Author contributions

Conceptualization: M.V., S.S., and P.P.D.F.; methodology: I.S.L., G.G., C.I., G.Se., S.C., S.F., F.B., S.Pe., G.B., D.T., M.G.M., A.D., and M.V.; formal analysis: D.D. (Figs. 1b, d, 10c, d; Supplementary Tables 1–3), S.C. (Figs. 1b, d, 6d, 9d, 10b; Supplementary Fig. 8D and Supplementary Tables 1–3), S.F. (Figs. 9f, 10b–d, Supplementary Tables 1–3), M.V. (Figs. 1b, d, 9c, d, 10a, b; Supplementary Table 2); investigation: I.S.L. (Figs. 2a, c, d, f, 4f, 5d, 6a–d, 7g, 8a–g; Supplementary Figs. 3A–C, 5F, 6A–E, 7A–C, 8A, 8C, D, 9A, B, 10A), G.G. (Figs. 2b–d, 3a, b, e, 4b–e, 5a, b, f, g, 6 a, 7a–f; Supplementary Figs 2A, 3B, 3E, 4B, C, 9C), C.I. (Fig. 2f, g, 4 a, 5 c; Supplementary Figs. 2C–E, 3D, 4A, 5A, B, 8B), C.T. (Figs. 3a, b, 9a, b, e, f, 10b; Supplementary Figs. 3E, 10B, C), G.C. (Fig. 5c; Supplementary Fig 5E), G.Se. (Fig. 1b; Supplementary Fig. 5C, D), S.Pi (Supplementary Fig. 2A, B, Supplementary Table 2), M.V. (Figs. 1b, d, 9c, d, 10a, b; Supplementary Figs. 2A, B, 10C, Supplementary Table 2), G.B. (Figs. 1c, 9b–e, 10a, b; Supplementary Figs. 1A, B, 10C, Supplementary Table 1), and A.D. (Figs. 2e, 5e); resources: G.Sc., G.V., and B.K.P.; data curation, F.B., D.D., S.C., and S.F.; writing—original draft, S.S., M.V., S.Pe., and P.P.D.F; writing—review and editing, I.S.L., G.G., C.I., S.C., G.B., G.V., D.D., D.T., M.G.M., G.Sc., S.Pe., S.S., M.V., and P.P.D.F.; supervision, S.S., M.V., and P.P.D.F.; funding acquisition, G.Sc., A.D., S.Pe., D.T., S.S., M.V., and P.P.D.F.

## Competing interests

The authors declare no competing interests.
