## [Peer Review File · Nature Communications]

Reviewers' comments:

Reviewer #1 (Remarks to the Author); expert in EMT and TGFb:

EPN3 is a protein involved in endocytosis and implicated in cell migration during wound healing. In the present study, the authors report that epsin 3 (EPN3) is often overexpressed in breast cancer and promotes cancer progression through induction of partial EMT. The process is mediated through accelerated E-cadherin internalization and resultant activation of beta-catenin/TCF-4 transcription pathway. Thus far, the mechanism is novel and interesting. In addition, they obtained data that support the role of EPN3 during cancer progression, utilizing patients' specimens. However, one point that makes this paper complicated is cooperation between EPN3 and TGF-beta. Some of the effects of EPN3 overexpression appear to be dependent on TGF-beta signaling, resulting in something like a chicken-or-egg situation. The initial event of the whole story is not as clear as shown in Figure 5E.

Major points:

1) As illustrated in Figure 5E, the authors think that accelerated ECAD internalization is the most upstream among events induced by EPN3 overexpression. However, the situation appears to be more complicated, because TGF-beta signaling may be involved in this process. In Figure 4A and B, TGF-beta-induced ECAD internalization was enhanced by EPN3. In Figure S4G, TGF-beta promoted interaction between EPN3 and ECAD. Thus, it appears likely that EPN3 is a downstream effector of TGF-beta during TGF-beta-induced ECAD internalization. The authors should examine if accelerated ECAD internalization induced by EPN3 overexpression is dependent on TGF-beta signaling or not. The result might explain why the endocytosis-promoting action of EPN3 is selective for E-cadherin (ECAD).

2) Similarly, beta-catenin/TCF4 is depicted as upstream of enhanced TGF-beta signaling. This may be misleading because some of the partial EMT phenotypes induced by EPN3 overexpression (induction of mesenchymal markers) is dependent not only on TCF4 but also TGF-beta signaling. Thus, the beta-catenin/TCF4 system is likely to cooperate with TGF-beta signaling.

3) Figure 4D shows that EPN3 overexpression resulted in sustained activation of TGF-beta signaling. Together with the data in Figure 5AB that show up-regulation of TGF-beta ligands by EPN3 overexpression, the authors concluded that TGF-beta-dependent autocrine loop is required for the sustained EMT phenotypes. However, contribution of the autocrine loop here was not satisfactorily demonstrated. To demonstrate this, Figure 4D should have included samples with no external TGF-beta as well as those with a TGF-beta type I receptor kinase inhibitor. In addition, autoinduction loop

of TGF-beta has been already well known (Kim et al., J Biol. Chem. 264, 7041-7045, 1989 and other papers). Previous works should be cited.

4) Another point concerning Figure 4D is that the current data do not exclude the possibility that EPN3 overexpression inhibited some negative feedback regulators of TGF-beta signaling. For example, TMEPAI is a membrane-bound negative feedback regulator of TGF-beta signaling (Watanabe et al., Mol Cell, 37, 123-134, 2010). Its membrane localization might be affected by EPN3 overexpression. Such possibility should be discussed (This could be the initial event induced by EPN3 overexpression).

5) In Figure S3B, EPN3 overexpression resulted in induction of EMT transcription factors, ZEB1, Snail and Twist, but did not affect ECAD mRNA expression. All of these EMT transcription factors are known to transcriptionally down-regulate ECAD expression. This is confusing to readers and needs to be discussed.

6) In Figure 5C, EPN3 induces mesenchymal markers, depending on TbRI and TCF4. Is induction of EMT-TFs (ZEB1, Snail, Twist) also dependent on both TCF4 and TbRI?

7) In Figure 5D, EPN3 upregulated expression of TGF-beta, NCAD and VIM even after knockdown of TbRI or TCF4. Does this suggest that EPN3 induces partial EMT independently of TGF-beta signaling or TCF4 to some extent?

8) The authors first observed that EPN3 overexpression resulted in ECAD downregulation without accompanying decrease of ECAD mRNA. If EPN3 overexpression establishes TGF-beta autocrine loop, ECAD downregulation would involve transcriptional repression through induction of EMT-TFs. One possibility is that the authors used MCF10A cells, which are not so often used in studies of TGF-beta-induced EMT. In addition, it is not appropriate to draw conclusions from experiments using only one cell type. I recommend the authors to repeat some of the important experiments using NMuMG, A549 or other cell lines that are commonly used in studies of TGF-beta-induced EMT.

Minor points:

9) In Figure 4, "AW" should be explained in the legend.

10) In Figure S6B, cooperation between EPN3 and TGF-beta is not clear, especially for VIM expression.

11) Figure 7A: Why was NCAD expression not altered by knockdown of EPN3?

12) Figure 7C: It is difficult to local up/down-regulation of ECAD and NCAD. Images with higher magnification would be helpful.

Reviewer #2 (Remarks to the Author); expert in breast cancer and stem cells:

This is well written manuscript that describes a role for Epsin 3 (EPN3) protein in the development of invasive properties in premalignant breast cancer cells. The experiments are thorough with some exceptions (noted below) and describe a previously unappreciated role for EPN3 in breast cancer progression towards the development of a metastatic disease.

Although the role of EPN1 in tumour invasion and progression has been described before, I agree with the authors that the role of EPN3 in breast cancer progression has remained unexplored and therefore this manuscript does provide novel and new information that might be relevant to solid tumours other than breast cancer.

However, I have some significant concerns that I believe lessens the impact and significance of the findings reported in this manuscript.

Major concerns:

The manuscript uses breast cell lines only and its impact could be strengthened by using primary human breast cells as controls and primary human breast cancer cells. MCF-10A, although not malignant, they do not recapitulate the primary normal breast epithelial cells that consist of ER+ and ER- luminal cells along with alpha smooth muscle actin positive myoepithelial cells. Since the majority of the signaling data is obtained from the ER- MCF10A cells the impact and expression EPN3 on the ER+ luminal and the basal myoepithelial cells remain unexplored. This is a significant limitation of the study. Also, considerations could have been given to the use of mouse breast cancer model that at an early stage show in situ disease (e.g. Her2/Neu model).

Fig 2. The quantification of the acini formation (regular vs irregular acini) is not described. How many acini structures were considered? These experiments should be repeated using primary human breast epithelial cells that include both luminal and myoepithelial cells. This is particularly important when considering acini formation and bi-layered structure of the breast organoids which is not possible when using MCF-10A.

Fig. 3B FACS data should be gated on live, single cells. Cells should first be gated based on Forward Scatter (FSC) and the side scatter (SSC) profile (to eliminate cells with high FSC and/or SSC) to exclude cell aggregates and then live cells should be gated based on PI (or other fluorescent cell viability dyes) exclusion. Then expression of ECAD can be examined in these live single cells.

Fig 3F needs to be quantified with 3 independent Western Blots and show statistical significance.

Polarization studies are done in 3D matrigles which is very appropriate. However, the internalization of the EPN3 and TCF are examined in MCF-10A cells grown on glass slides. Although it is appreciated that glass surface is more conducive to immunofluorescent studies, yet it has now been convincingly shown that cells grown in 3D behave differently than cells grown in 2D with respect to cell signaling and protein localization within the cells. To be consistent with the polarization data shown in Fig. 3, the internalization of beta catenin and EPN3 should also be performed in 3D matrigles.

Fig 6. Shows cancer stem cell marker expression (CD44 and CD24) in MCF-10A and MCF-10A-EPN1 cells. However, the gates shown in the FACS plots are not drawn appropriately. MCF10A cells are breast epithelial cells obtained from a woman's breast tissue with fibrocystic disease. The cells that did grow out in 2D cultures are luminal in nature and therefore it comes as very little surprise that they are nearly 100% positive for CD24 (a luminal cell marker in breast epithelial cells) expression irrespective of EPN3 or TWIST expression. Also, as shown in Fig. 6A all MCF10A cells express high levels of CD44. Clearly expression of CD44 and CD24 in MCF10A cells is not a marker for cancer stem cells or their properties. The biological relevance and significance of marginal changes in already highly expressed CD44 is not immediately clear. This data could be better presented as histograms.

Once again, it is important to be consistent and grow all cells in 3D Matrigel cultures.

Overall, the data presented in support of a role for EPN3 in cancer stem cell phenotype/properties is not strong and is very descriptive in nature. In vivo animal experiments are required to measure cancer stem cell properties.

Fig 7. The data shown in support of the notion that EPN3 is upregulated at the "invasive front" and required for tumour progression is not strong and is speculative.

Minor comments:

Figure 2F is only based on 2 independent experiments. All experiments should be based on at least 3 independent experiments.

Need nuclear staining (DAPI) for Fig. 3E

What is meant by “AW” in Fig. 4A?

Fig. 5B is based on only 2 biological replicates and needs to be repeated to 3 or 4 biological replicates.

Fig. 5C needs to be repeated and quantified.

Fig 7 legend is not clear at all and does not convey the data that is being shown. How was the data shown in Fig 7D generated? How many sections were examined? Where the xenografts the same size and weight between control and the experiment arm? How was this taken into consideration?

Reviewer #3 (Remarks to the Author); expert in endocytosis and cell signalling:

Manuscript NCOMMS-19-14370-T.

The work that is presented in the current manuscript has been performed by the teams of Di Fiore and Sigismund. The findings focus on the interplay between Epsin-3 (EPN3) and TGFbeta-based signalling, with a specific emphasis on endocytic uptake. The study is very original in its outlay, and the manuscript well written for a general readership in the life sciences. It is very timely to deepen the mechanistic link between endocytic trafficking and EMT, as it is proposed in the current manuscript. Di Fiore and Sigismund are perfectly positioned with exquisite expertise in endocytosis and carcinogenesis.

I would have a few comments on the endocytosis-related aspects, which clearly are one of the several highlights of the current study. These comments are listed as they appear in the manuscript.

The E-cadherin (ECAD) endocytic uptake kinetics as shown in Figure 3C is very slow. Is this a constitutive turnover? If so, what would be its physiological function? It would be helpful for the general readership if this aspect could be further discussed.

Related to this point: It is not clear from the materials and methods section whether ECAD endocytosis was followed by primary anti-ECAD antibody only, or after formation of a complex between primary and secondary antibodies. If the latter was the case, it should be explained how the authors can exclude extensive crosslinking, which would likely have an influence of the uptake kinetics (and possibly pathways). If the former was the case, the protocols fails to mention the labelling step with secondary antibodies after uptake of the primary ones.

The specific effect of EPN3 on ECAD endocytosis is quite interesting. It would again be of help to the general readership if the authors could speculate on how this specificity might be achieved.

On page 8, the authors mention that “EPN3 appears to work as an endocytic adaptor linking ECAD adherens junctions to clathrin”. Does EPN3 still induce ECAD endocytosis when the clathrin pathway is inhibited?

Depletion of TCF4 in MCF10A-EPN3 cells reverts the EPN3-induced expression of 2 EMT targets, VIM and NCAD. Is this reversion also observed when the clathrin pathway is inhibited?

Does the TGF-beta mediated stimulation of ECAD endocytosis depend on clathrin?

EPN3 has a massive effect on the faction of CD44^{high}/CD24^{low} cells in the MCF10A cell population, which is very exciting. Would there be a way to establish a link between this effect and endocytic uptake? For example, does this effect depend on clathrin pathway function, and/or the endocytosis of CD44 or ECAD?

Point-by-point reply to the Reviewers' comments:

Reviewer #1

EPN3 is a protein involved in endocytosis and implicated in cell migration during wound healing. In the present study, the authors report that epsin 3 (EPN3) is often overexpressed in breast cancer and promotes cancer progression through induction of partial EMT. The process is mediated through accelerated E-cadherin internalization and resultant activation of beta-catenin/TCF-4 transcription pathway. Thus far, the mechanism is novel and interesting. In addition, they obtained data that support the role of EPN3 during cancer progression, utilizing patients' specimens.

R: We thank the Reviewer for his/her positive comments.

However, one point that makes this paper complicated is cooperation between EPN3 and TGF-beta. Some of the effects of EPN3 overexpression appear to be dependent on TGF-beta signaling, resulting in something like a chicken-or-egg situation. The initial event of the whole story is not as clear as shown in Figure 5E.

R: Agree. We have performed additional experiments to address this issue (see answer to Major point 1).

Major points:

1) As illustrated in Figure 5E, the authors think that accelerated ECAD internalization is the most upstream among events induced by EPN3 overexpression. However, the situation appears to be more complicated, because TGF-beta signaling may be involved in this process. In Figure 4A and B, TGF-beta-induced ECAD internalization was enhanced by EPN3. In Figure S4G, TGF-beta promoted interaction between EPN3 and ECAD. Thus, it appears likely that EPN3 is a downstream effector of TGF-beta during TGF-beta-induced ECAD internalization. The authors should examine if accelerated ECAD internalization induced by EPN3 overexpression is dependent on TGF-beta signaling or not. The result might explain why the endocytosis-promoting action of EPN3 is selective for E-cadherin (ECAD).

2) Similarly, beta-catenin/TCF4 is depicted as upstream of enhanced TGF-beta signaling. This may be misleading because some of the partial EMT phenotypes induced by EPN3 overexpression (induction of mesenchymal markers) is dependent not only on TCF4 but also TGF-beta signaling. Thus, the beta-catenin/TCF4 system is likely to cooperate with TGF-beta signaling.

R: Agree. As the reviewer correctly pointed out, we had no definitive proof that accelerated ECAD internalization induced by EPN3 was the initial event. Indeed, EPN3 overexpression, in otherwise unperturbed cells, at steady state established a feedback loop dependent on TGF-beta, which acts both downstream and upstream the ECAD-EPN3 axis.

To address this point, we generated a doxycycline-inducible EPN3 overexpression system (see new Fig. 6) that allowed us to define the temporal order of the events occurring after EPN3 induction. We found that, at 6h after the induction of EPN3 overexpression, ECAD endocytosis was accelerated (Fig. 6D, E), cell-to-cell junctions were rearranged (Fig. 6D) and active beta-catenin was translocated to the nucleus (Fig. 6F). These events occurred in the absence of TGF-beta ligand/receptor upregulation (Fig. 6C), and thus before the establishment of the TGF-beta-dependent loop. The loop started to be activated later, at 24 h after EPN3 induction, when upregulation of TGF-beta-1 ligand and TGF-beta-R, and of early EMT markers, such as SNAIL and FN1 (Fig. 6C), was observed, and a more dramatic morphological change was visible (Fig. 6B).

Of note, once the loop is established, the beta-catenin/TCF4 system likely cooperates with TGF-beta signaling, as pointed out by the reviewer and supported by the evidence that the TGF-beta-dependent transcriptional events are also in part dependent on beta-catenin/TCF4 (*i.e.* TCF4 KD partially reverted TGF-beta-induced transcriptional program, see Fig.5D and S7D).

3) Figure 4D shows that EPN3 overexpression resulted in sustained activation of TGF-beta signaling. Together with the data in Figure 5AB that show up-regulation of TGF-beta ligands by EPN3 overexpression, the authors concluded that TGF-beta-dependent autocrine loop is required for the sustained EMT phenotypes. However, contribution of the autocrine loop here was not satisfactorily demonstrated. To demonstrate this, Figure 4D should have included samples with no external TGF-beta as well as those with a TGF-beta type I receptor kinase inhibitor. In addition, autoinduction loop of TGF-beta has been already well known (Kim et al., J Biol. Chem. 264, 7041-7045, 1989 and other papers). Previous works should be cited.

R: Agree. We apologize for the lack of clarity in the previous version of the manuscript. In Fig. 4D and in Fig. S4H, the (-) samples are without external TGF-beta. In addition, we have now included a new experiment in which we treated MCF10A-EPN3 cells with LY2109761 (a TGF-beta type I receptor kinase inhibitor) also in the absence of TGF-beta stimulation (new Fig. S8A,B). This reverted the EPN3-dependent upregulation of EMT markers, TGFβ1, NCAD, VIM, FN1, SNAIL, SLUG and TWIST. The reversion was not complete suggesting that there might be some EPN3-specific effects, which are independent of TGF-beta, as also pointed by the reviewer (see point 7 of this reviewer and the relative reply).

We have also discussed and cited previous work on the establishment of TGF-beta loops and EMT (page 20 of main text).

4) Another point concerning Figure 4D is that the current data do not exclude the possibility that EPN3 overexpression inhibited some negative feedback regulators of TGF-beta signaling. For example, TMEM41 is a membrane-bound negative feedback regulator of TGF-beta signaling (Watanabe et al., Mol Cell, 37, 123-134, 2010). Its membrane localization might be affected by EPN3 overexpression. Such possibility should be discussed (This could be the initial event induced by EPN3 overexpression).

R: Agree. We cannot exclude the possibility that EPN3, in addition to regulating ECAD endocytosis, is also inhibiting some negative feedback regulators of TGF-beta signaling, as the one suggested by the reviewer. We have now discussed this possibility in the Discussion section (page 20).

5) In Figure S3B, EPN3 overexpression resulted in induction of EMT transcription factors, ZEB1, Snail and Twist, but did not affect ECAD mRNA expression. All of these EMT transcription factors are known to transcriptionally down-regulate ECAD expression. This is confusing to readers and needs to be discussed.

R: Agree. Our data suggest that there are different degrees of EMT, in agreement with literature (Aiello NM *et al.*, 2018; Jolly MK *et al.*, 2018; Pastushenko I *et al.*, 2018). In particular, TWIST is able to induce a more advanced EMT as compared to EPN3 (Fig. 2B-F and S3B-D). Indeed, in the case of TWIST, we observed a stronger induction of the EMT transcription factor ZEB1 (in addition to TWIST itself, Fig. 2D and S3B), which resulted in a stronger upregulation of mesenchymal markers (NCAD, VIM and FN1), as compared to EPN3, and might explain the transcriptional repression of ECAD mRNA. In the case of EPN3, the upregulation of ZEB1 and TWIST was lower and did not result in the transcriptional downregulation of ECAD. This is reflected also morphologically, since EPN3 cells presented a less pronounced mesenchymal

phenotype, in which cell-to-cell junctions are disorganized but not completely destroyed (Fig. 2B and S4A). Thus, the EPN3 phenotype seems to better be described as the so-called ‘partial EMT phenotype’, where mesenchymal and epithelial markers coexist (Aiello NM *et al*, 2018; Jolly MK *et al*, 2018; Pastushenko I *et al*, 2018).

Importantly, treatment with TGF-beta enhanced the EPN3-induced EMT (new Fig. 4D and S4H), by further upregulating the EMT transcription factors ZEB1 and SNAIL (new Fig. S4H and S7D), finally leading to ECAD transcriptional downmodulation (see new Fig. S4H). We have now discussed this issue on page 18 of the main text.

6) *In Figure 5C, EPN3 induces mesenchymal markers, depending on TbRI and TCF4. Is induction of EMT-TFs (ZEB1, Snail, Twist) also dependent on both TCF4 and TbRI?*

R: Agree. We have included these data in new Fig. S7D. The induction of ZEB and SNAIL is dependent, to some extent (see also point 7 of this reviewer and the relative answer), on TGF-beta and/or TCF4 and, indeed, they were upregulated by TGF-beta stimulation, and their upregulation was impaired upon TGFβR1 and TCF4 KD. On the contrary, the upregulation of TWIST was not reverted by TGF-beta and/or TCF4 KD and TWIST mRNA levels were not regulated by TGF-beta stimulation. These results again suggest the possibility that some of the EPN3-induced phenotypes are independent of TGF-beta and/or TCF4.

7) *In Figure 5D, EPN3 upregulated expression of TGF-beta, NCAD and VIM even after knockdown of TbRI or TCF4. Does this suggests that EPN3 induces partial EMT independently of TGF-beta signaling or TCF4 to some extent?*

R: We agree with the reviewer. The results in Fig. 5D and the new data with TGF-beta type I receptor kinase inhibitor in new Fig. S8A,B confirm that there is a partial reversion of EPN3 induced EMT upon TGF-beta and/or TCF4 KD. Although we cannot exclude that this is due to incomplete KD, this data seems to argue that EPN3-induced partial EMT is - at least to some extent - independent of TGF-beta signaling or TCF4. We have discussed this point at page 12 and at page 20 of the main text.

8) *The authors first observed that EPN3 overexpression resulted in ECAD downregulation without accompanying decrease of ECAD mRNA. If EPN3 overexpression establishes TGF-beta autocrine loop, ECAD downregulation would involve transcriptional repression through induction of EMT-TFs. One possibility is that the authors used MCF10A cells, which are not so often used in studies of TGF-beta-induced EMT. In addition, it is not appropriate to draw conclusions from experiments using only one cell type. I recommend the authors to repeat some of the important experiments using NMuMG, A549 or other cell lines that are commonly used in studies of TGF-beta-induced EMT.*

R: Agree. In the revised manuscript, we have included experiments performed in another normal breast epithelial cell of different derivation compared to MCF10A, *i.e.* MCF12A (new Fig. S5). In addition, data on another mammary epithelial cancer cell line HCC1569 were already present in the previous version of the manuscript (new Fig. S6). In both cases, we observed that EPN3 overexpression was able to induce changes in cell morphology resembling EMT phenotypic alterations, and to cause the upregulation of mesenchymal markers, although to a different extent and with a different pattern, as compared to MCF10A cells. In particular:

1. MCF12A normal breast cells resembled closely the MCF10A system. In this cells, EPN3 caused an elongated mesenchymal-like phenotype (Fig. S5A), the upregulation of the mesenchymal markers NCAD and VIM, at the protein level (Fig. S5B), and FN1, SNAIL, TWIST, ZEB1, and TGF-beta receptor 2, at the mRNA level (Fig. S5C). ECAD protein levels were slightly decreased,

while its mRNA levels were not significantly downregulated, in agreement with what we have observed in MCF10A cells. Importantly, TWIST caused a more advanced EMT as compared to EPN3, confirming that EPN3 phenotype might represent a partial EMT. Moreover, stimulation with TGF β further increased the EPN3-induced partial EMT phenotype, causing the upregulation of NCAD (Fig. S5D), the upregulation and the sustained induction of SNAIL and pSMAD2 (Fig. S5D), the upregulation of FN1 mRNA levels (Fig. S5E, bottom) and the downregulation of ECAD at both protein (Fig. S5D) and mRNA levels (this latter occurring in EPN3 overexpressing, but not in control cells, as in MCF10A cells, Fig. S5E, top).

2. Similarly, EPN3 overexpression in the BC cell line HCC1569 also induced a partial EMT phenotype (although to a lower extent as compared to MCF10A cells, as these cells present already quite high levels of VIM), showing an increase in NCAD, VIM and TGF-beta 1 by IB and/or RT-qPCR (Fig. S6A and S6B). In this cell model system, we could also confirm some cooperation between EPN3 overexpression and TGF-beta signaling in the induction of EMT (Fig. S6C).

Altogether, these data suggest that, while EPN3 overexpression is able to induce a partial EMT phenotype and to cooperate with TGF-beta signaling in different breast cell lines with a similar impact.

Finally, we have performed experiments also in A549 lung cancer epithelial cells, as suggested by the reviewer. Since these cells represent a completely different cell model system, with respect to mammary cells, we have not included the data in the manuscript and we have attached the relative figure to this letter (see attached Fig. 1). We are ready, however, to add these data to the study, if the reviewer thinks that this is necessary.

Figure 1 for reviewers. EPN3 overexpression in A459 cells. (A) A549 cells were infected with pLVX empty vector (Ctr), pLVX vector expressing EPN3 or pBABE vector expressing TWIST, and analyzed by IB analysis for the expression of EPN3, TWIST, and EMT markers in comparison to MCF10A-Ctr or -EPN3 cells. GAPDH was used as a loading control. MW markers are shown on the left. (B) Representative phase contrast microscopy pictures of A549 cells as described in (A). Bar, 250 μ m. (C) RT-qPCR of the indicated EMT markers in A549 cells as described in (A). Data were normalized on the average expression of 18S-ACTB-GAPDH and results are reported as relative fold change compared to A549-CTR \pm SD of independent experiments (n=5 for NCAD and VIM; n=2 for SLUG; n=3 for TGF β 2). P-value was calculated with Student's t test, two tailed. *, <0.05; ** <0.01. (D) A549-CTR and -EPN3 cells were stimulated with 5 ng/ml of TGF β for the indicated time points and subjected to IB analysis with the indicated antibodies. GAPDH

was used as a loading control. MW markers are shown on the left. (E) The same samples described in (D) were analyzed for FN1 and ECAD mRNA levels by RT-qPCR. Data were normalized on the expression of 18S and results are reported as relative mRNA expression compared to A549-Ctr (not stimulated) \pm SD of two independent experiments, each performed in technical triplicates. P-value was calculated with Each Pair Student's t-test. *, <0.05; **, <0.01***, <0.001.

In the A549 cell model system, EPN3 overexpression was able to induce changes in cell morphology resembling a partial EMT, and to cause the upregulation of mesenchymal markers (see attached Fig. 1 A-C) The major difference compared to the breast cell lines is that is that TGF-beta treatment caused an almost immediate and complete downregulation of ECAD, both at protein and mRNA levels, even in the absence of EPN3 overexpression (already after 1 day of treatment, see attached Fig. 1D, E right). No major synergy between EPN3 and TGF-beta was observed for ECAD downregulation (see attached Fig. 1D, E right), for the upregulation of NCAD and VIM (see attached Fig. 1D) and for the induction of FN1 mRNA level (see attached Fig. 1E left).

One possible explanation for this difference between A549 cells and the mammary cell lines is that A549 are exquisitely sensitive to TGF-beta, thus making the putative cooperation with EPN3 difficult to score. Further addressing of this point would require careful titration of sub-optimal doses of TGF-beta (with and without EPN3): an issue of interest but probably outside the scope of the present study, which is focused on breast cancer.

Minor points:

9) *In Figure 4, "AW" should be explained in the legend.*

R: Agree. "AW" stands for "acid wash". We have included this information in the legend to Figure 4A.

10) *In Figure S6B, cooperation between EPN3 and TGF-beta is not clear, especially for VIM expression.*

R: We agree with the reviewer. In the HCC1569 cell model system, we could observe a cooperation between EPN3 and TGF-beta in the case of NCAD expression at all the analyzed time points. In the case of VIM, some cooperation could be observed only after 7 days of TGF-beta treatment. We have now included a lower exposure of the western blot, where an increase in VIM expression upon EPN3 and TGF-beta treatment is visible at day 7 (new Figure S6C).

11) *Figure 7A: Why was NCAD expression not altered by knockdown of EPN3?*

R: What emerges from the results obtained in Figure 7A (now Fig. 8A) is that the effects of EPN3 KD on DCIS system are much stronger when cells are grown in the *in vivo* 3D context as compared to the *in vitro* 2D culture (e.g. downregulation of NCAD and VIM, and upregulation of ECAD). This might have a dual explanation: i) first, while in DCIS grown in 2D EPN3 levels are not upregulated (and are very close to the physiological level), in the *in vivo* situation and, in particular, during the transition from DCIS to IBC, EPN3 levels are upregulated and correlate with the acquisition of EMT traits; ii) second, in the *in vivo* context there are additional factors that might contribute to the EPN3 phenotype, e.g. cell-matrix interaction, interplay with the stroma, release of soluble factors (e.g. TGF-beta itself or others). We have highlighted this point at page 15 of the revised manuscript.

12) *Figure 7C: It is difficult to local up/down-regulation of ECAD and NCAD. Images at higher magnification would be helpful.*

R: Agree. We have included pictures at a higher magnification.

Reviewer #2

This is well written manuscript that describes a role for Epsin 3 (EPN3) protein in the development of invasive properties in premalignant breast cancer cells. The experiments are thorough with some exceptions (noted below) and describe a previously unappreciated role for EPN3 in breast cancer progression towards the development of a metastatic disease.

Although the role of EPN1 in tumour invasion and progression has been described before, I agree with the authors that the role of EPN3 in breast cancer progression has remained unexplored and therefore this manuscript does provide novel and new information that might be relevant to solid tumours other than breast cancer.

However, I have some significant concerns that I believe lessens the impact and significance of the findings reported in this manuscript.

R: We appreciate the positive comments of the reviewer. In the new version of the manuscript we have added several experiments to address his/her concerns.

Major concerns:

The manuscript uses breast cell lines only and its impact could be strengthened by using primary human breast cells as controls and primary human breast cancer cells. MCF-10A, although not malignant, they do not recapitulate the primary normal breast epithelial cells that consist of ER+ and ER- luminal cells along with alpha smooth muscle actin positive myoepithelial cells. Since the majority of the signaling data is obtained from the ER- MCF10A cells the impact and expression EPN3 on the ER+ luminal and the basal myoepithelial cells remain unexplored. This is a significant limitation of the study. Also, considerations could have been given to the use of mouse breast cancer model that at an early stage show in situ disease (e.g. Her2/Neu model).

R: Agree. This is a relevant point. To address the impact of EPN3 overexpression in a more physiological setting, we took advantage of a conditional knock-in mouse model that we have recently generated, in which the human *EPN3* gene under the *UBC* promoter has been inserted in the *ROSA26* locus with a STOP cassette flanked by two LoxP-sites (referred as *EPN3-KI*). Treatment with CRE recombinase allows for EPN3 overexpression. We are currently crossing these mice with mice harboring CRE expression under a mammary-specific promoter, to perform *in vivo* studies. These experiments will require time and efforts that go beyond the scope of the present study.

However, for the purpose of the present study, we have exploited these mice to obtain primary mammary epithelial organoids, where we can induce EPN3 overexpression *ex vivo* (by CRE treatment) and study their growth in Matrigel. Importantly, these organoids resemble the double-layered mammary acini, with both myoepithelial (K14-positive) and luminal (K8-positive) cells (new Fig. S3E). Upon induction of EPN3 overexpression, organoids acquired a strong invasive phenotype (new Fig. 2H), they lost their organized bi-layered structure and displayed invasive cells delaminating from the organoid core. These cells showed loss of epithelial identity (*i.e.* ECAD, K5 and K8), and acquired the expression of the mesenchymal marker NCAD (new Fig. 2I), recapitulating the EPN3-induced EMT observed in MCF10A cells.

In Fig 2. The quantification of the acini formation (regular vs irregular acini) is not described. How many acini structures were considered? These experiments should be repeated using primary human breast epithelial cells that include both luminal and myoepithelial cells. This is particularly important when considering acini formation and bi-layered structure of the breast organoids which is not possible when using MCF-10A.

R: Agree. Concerning MCF10A-derived organoids (Fig. 2F), we apologize for the lack of sufficient explanations, in the previous version of the manuscript. We have now described how regular vs. irregular acini were quantified (see legend to Fig. 2F and Methods, section “Matrigel MCF10A morphogenetic assay”). We have also included an additional quantification of the number of acini, as we scored an impact of EPN3 and TWIST not only on the morphology at later time points but also on the growth of the organoids at early time points (new Fig. S3C).

Concerning the use of *primary human breast epithelial cells that include both luminal and myoepithelial cells*, given the difficulties in obtaining human primary samples, we have performed experiments with organoids derived from mouse primary breast epithelial cells, which resembled the bi-layered structure of the breast acini (see previous point).

Fig. 3B FACS data should be gated on live, single cells. Cells should first be gated based on Forward Scatter (FSC) and the side scatter (SSC) profile (to eliminate cells with high FSC and/or SSC) to exclude cell aggregates and then live cells should be gated based on PI (or other fluorescent cell viability dyes) exclusion. Then expression of ECAD can be examined in these live single cells.

R: All our FACS experiments are gated on single cells, that have been previously fixed. Briefly, cells were gated first on Side Scatter (SSC)-Area and Side Scatter (SSC)-Height to select for single cells and exclude aggregates, then were gated on Forward Scatter (FSC) and Side Scatter (SSC) profile to eliminate cells with high FSC and/or SSC; finally, cells were gated based on ECAD expression. Since we have used this assay to measure in parallel ECAD surface levels and ECAD internalization rate, which requires an acid wash treatment, it was not possible to leave the cells alive but we had to fix them immediately after the treatment.

In response to the reviewer’s point, we have compared (during the set-up phase of our protocol) live vs. fixed cells, and we did not observe any significant difference (attached Fig. 2). These results are now mentioned in our Methods section (section “PM ECAD intensity by FACS”).

Figure 2 for reviewers. Comparison of ECAD staining and FACS analyses in MCF10A-Ctr cells prior or after fixation. (A) MCF10A-Ctr cells were subjected to EGF-starvation for 3 hrs, then stained with anti-ECAD antibody (Abcam, 2 µg/ml) for 1 h at 4°C, followed by incubation with secondary antibody (Alexa-488, 5 µg/ml) for 30 min at 4°C. Cells were then washed twice with PBS and 0.25% trypsin was added for 15-20 min at 37°C. Cells were recovered in medium in a 15 ml Falcon, washed once with PBS, re-suspended in 500 µl PBS and fixed by adding 500 µl of formaldehyde 2% for 15 min on ice or left untreated (not fixed). Subsequently, cells were centrifuged 335g for 5 min and re-suspended in PBS with EDTA 2 mM before analysis with the FACS Celesta (BD). Samples stained only with secondary antibody were used as negative control. Analysis was performed using the FlowJo Software (LLC). Briefly, cell doublets and clumps were excluded through the SSC Area (Side Scatter Area) over the height peak parameter. Cells with correct morphology were then selected through the FSC Area (Forward Scatter Area) over SSC Area parameter. Finally, ECAD mean fluorescence intensity of the population was used to compare the different samples. **(B)** % of ECAD positive cells of the indicated samples as described in (A). **(C)** Relative ECAD fluorescence intensity of the indicated samples as described in (A).

Fig 3F needs to be quantified with 3 independent Western Blots and show statistical significance.

R. Agree. We have quantified the effects of both TCF4 KD (Fig. 3F and 5C) and TGF-beta receptor KD (Fig. 5C) on ECAD, NCAD and VIM protein levels. Mean and statistics are shown in new Fig. S7C and are based on at least 5 independent experiments (n=5 for TGF-beta R1 KD samples; n=6 for TCF4 KD samples).

Polarization studies are done in 3D matrigels which is very appropriate. However, the internalization of the EPN3 and TCF are examined in MCF-10A cells grown on glass slides. Although it is appreciated that glass surface is more conducive to immunofluorescent studies, yet it has now been convincingly shown that cells grown in 3D behave differently than cells grown in 2D with respect to cell signaling and protein localization within the cells. To be consistent with the polarization data shown in Fig. 3, the internalization of beta catenin and EPN3 should also be performed 3D matrigels.

R. Agree. We presume that the reviewer is referring here to the monitoring of ECAD internalization and beta-catenin nuclear translocation in 3D Matrigel. We could not perform the experiment under the same protocol that we employed in 2D, *i.e.* by providing the ECAD antibody *in vivo* to the organoids. Under these conditions it was difficult to detect any internalization of ECAD. We suspect that the Ab cannot diffuse well enough in the organoid.

To circumvent the problem, we exploited a doxycycline inducible EPN3 expression system in MCF10A cells (see reply to Point 1-2 of Reviewer 1 and new Fig. 6) that allowed us to induce EPN3 overexpression (and, eventually, EPN3-induced ECAD internalization and beta-catenin translocation) in a specific timeframe after organoid formation. After 13 days of organoid growth in Matrigel, EPN3 overexpression was induced for 24 h. Then, organoids were fixed and stained for ECAD and active beta-catenin. With this system, we managed to observe a rearrangement of ECAD cell-to-cell junctions and a translocation of active dephosphorylated beta-catenin in the nucleus within the organoid structures (new Fig. 6E). No changes in ECAD or beta-catenin localization were observed in doxycycline-treated control cells.

Fig 6. Shows cancer stem cell marker expression (CD44 and CD24) in MCF-10A and MCF-10A-EPN1 cells. However, the gates shown in the FACS plots are not drawn appropriately. MCF10A cells are breast epithelial cells obtained from a woman's breast tissue with fibrocystic disease. The cells that did grow out in 2D cultures are luminal in nature and therefore it comes as very little surprise that they are nearly 100% positive for CD24 (a luminal cell marker in breast epithelial cells) expression irrespective of EPN3 or TWIST expression. Also, as shown in Fig. 6A all MCF10A cells express high levels of CD44. Clearly expression of CD44 and CD24 in MCF10A cells is not a

marker for cancer stem cells or their properties. The biological relevance and significance of marginal changes in already highly expressed of CD44 is not immediately clear. This data could be better presented as histograms.

Once again, it is important to be consistent and grow all cells in 3D Matrigel cultures. Overall, the data presented in support of a role for EPN3 in cancer stem cell phenotype/properties is not strong and is very descriptive in nature. In vivo animal experiments are required to measure cancer stem cell properties.

R: We have removed the sentences concerning the impact of EPN3 on the cancer stem cell compartment from the Abstract and the Discussion, and we have toned down our conclusions in the Result section (see pages 13-14 and 17). In addition, we have plotted data also as histograms, which showed the appearance of a second population of cells in the case of MCF10A-EPN3 and -TWIST for both CD44 (the second peak with a higher expressing population) and CD24 (the first peak with lower expressing population) (see new Fig. S9A).

Fig 7. The data shown in support of the notion that EPN3 is upregulated at the “invasive front” and required for tumour progression is not strong and is speculative.

R: In compliance with the reviewer’s criticism, we have toned down the relative statements and conclusions (page 15). We respectfully point out, however, that the upregulation of EPN3 at the invasive front is pretty evident and that the correlation between EPN3 expression and invasion is not merely speculative, as it is supported by molecular genetics in EPN3-silenced cells (Fig. 8F).

Minor comments:

Figure 2F is only based on 2 independent experiments. All experiments should be based on at least 3 independent experiments.

R: Agree. The new Figure 2F is now representative of 3 experiments.

Need nuclear staining (DAPI) for Fig. 3E

R: Agree. The DAPI staining is now included.

What is meant by “AW” in Fig. 4A?

R: We apologize for lack of explanation. “AW” stands for “acid wash”. We have included this information in the legend to Figure 4A.

Fig. 5B is based on only 2 biological replicates and needs to be repeated to 3 or 4 biological replicates.

R: Agree. The new Figure 5B is now representative of three biological replicates, each performed on the medium of at least 2 independent dishes, each of which was assayed in technical triplicates.

Fig. 5C needs to be repeated and quantified.

R: Agree. Quantification is now shown in new Fig. S7C and it is the mean of at least 5 independent experiments (n=5 for TGF-beta R1 KD samples; n=6 for TCF4 KD samples).

Fig 7 legend is not clear at all and does not convey the data that is being shown. How was the data shown in Fig 7D generated? How many sections were examined? Where the xenografts the same size and weight between control and the experiment arm? How was this taken into consideration?

R: Agree. We have now better explained in legend to new Fig. 8D (old Fig. 7D) the points raised by the reviewer.

Reviewer #3

The work that is presented in the current manuscript has been performed by the teams of Di Fiore and Sigismund. The findings focus on the interplay between Epsin-3 (EPN3) and TGFbeta-based signalling, with a specific emphasis on endocytic uptake. The study is very original in its outlay, and the manuscript well written for a general readership in the life sciences. It is very timely to deepen the mechanistic link between endocytic trafficking and EMT, as it is proposed in the current manuscript. Di Fiore and Sigismund are perfectly positioned with exquisite expertise in endocytosis and carcinogenesis.

R: We thank the reviewer for his/her appreciative comments.

1. I would have a few comments on the endocytosis-related aspects, which clearly are one of the several highlights of the current study. These comments are listed as they appear in the manuscript. The E-cadherin (ECAD) endocytic uptake kinetics as shown in Figure 3C is very slow. Is this a constitutive turnover? If so, what would be its physiological function? It would be helpful for the general readership if this aspect could be further discussed.

R: We thank the reviewer for highlighting this point. In Fig. 3C we followed the constitutive endocytosis of ECAD, as he/she correctly pointed out. At steady state, the slow, but continuous, endocytic uptake and recycling of ECAD is involved in adherens junction remodeling, a process that has been shown to be critical for the establishment of epithelial cell polarity, epithelial cell division and tissue integrity (Cadwell CM *et al.*, *Traffic* 2016; Bruser L *et al.*, *Cold Spring Harb Perspect Biol* 2017). Treatment with TGF-beta has been shown to increase the turnover of ECAD and to destabilize adherent junctions, a process that, in combination with the activation of the TGF-beta signaling cascade, finally leads to the EMT phenotypes (Corallino S *et al.*, *Front. Onc.* 2015). We have further discussed this issue in the text (page 19 of the Discussion).

2. Related to this point: It is not clear from the materials and methods section whether ECAD endocytosis was followed by primary anti-ECAD antibody only, or after formation of a complex between primary and secondary antibodies. If the latter was the case, it should be explained how the authors can exclude extensive crosslinking, which would likely have an influence of the uptake kinetics (and possibly pathways). If the former was the case, the protocols fails to mention the labelling step with secondary antibodies after uptake of the primary ones.

R: The reviewer is right. In the FACS-based assays, ECAD endocytosis was followed after the formation of a complex between primary and secondary antibodies. However, to validate the FACS result with another technology, we have now repeated the IF-based assays by staining cells *in vivo* with primary anti-ECAD antibody only (and performing the secondary antibody staining after fixation of cells, see new Methods, section “ECAD internalization assays by IF”). We have also included a quantitation of the IF (internalized ECAD signal relative to control sample), which showed results that are comparable to the FACS-based endocytic assay (new Fig. 4A).

3. The specific effect of EPN3 on ECAD endocytosis is quite interesting. It would again be of help to the general readership if the authors could speculate on how this specificity might be achieved.

R: This is an interesting point, on which we can only speculate, presently. The specificity is actually dual: a) EPN1, despite high homology with EPN3 does not seem to elicit any of the effects exerted by EPN3, b) EPN3 modulates ECAD internalization but not that of EGFR and TGF-beta receptor (with the obvious caveat that in this case “specificity” is limited to the analyzed receptors).

With regard to the first point, EPN3 shows a high degree of similarity with the other family members, EPN1 and EPN2, with more than 90% identity in the ENTH domain. All three members possess binding motifs for clathrin, AP2 and ubiquitin. The central and the C-terminal portions are the most divergent ones. Of note, EPN3 presents a specific sequence after the ubiquitin-binding motifs codified by a specific exon (Exon 5) conserved between human, mouse and rat, which is not present in the other two family members; thus, this region might contain some of the determinants of EPN3 specificity. In the future, it will be of interest to identify the “minimal” EPN3 region responsible for the effects, with the perspective of developing specific inhibitors. We have discussed this point in the main text at pages 20-21.

With regard to the second point, EPN3 has been shown to be involved in the endocytosis of some receptors, including the EGFR and TfR (Messa E *et al.*, 2014, Boucrot E *et al.*, 2012). In these instances, EPN3 appeared to have a redundant role with EPN1 and EPN3. Thus, the only conclusion that we can draw at present is that EPN3 has a non-redundant role in ECAD internalization. The molecular basis for this “specificity” might overlap (or not) with those discussed at the above point. We have further discussed these points in our revised discussion (page 19). A better understanding of the molecular mechanisms of EPN3-induced endocytosis of ECAD will be required to shed light on these issues: a question also raised by the reviewer, which we discuss below.

4. *On page 8, the authors mention that “EPN3 appears to work as an endocytic adaptor linking ECAD adherens junctions to clathrin”. Does EPN3 still induce ECAD endocytosis when the clathrin pathway is inhibited?*

Depletion of TCF4 in MCF10A-EPN3 cells reverts the EPN3-induced expression of 2 EMT targets, VIM and NCAD. Is this reversion also observed when the clathrin pathway is inhibited? Does the TGF-beta mediated stimulation of ECAD endocytosis depend on clathrin? EPN3 has a massive effect on the fraction of CD44^{high}/CD24^{low} cells in the MCF10A cell population, which is very exciting. Would there be a way to establish a link between this effect and endocytic uptake? For example, does this effect depend on clathrin pathway function, and/or the endocytosis of CD44 or ECAD?

R: We have put considerable effort in trying to address this point. We attached the data to this letter, as results, while technically very clear, are not conclusive and could only be “partially” interpreted. Thus, we are hesitant to include them in the paper.

The reviewer asks for experiments along three major lines:

- A). Is the ECAD internalization pathway activated by EPN3 a clathrin-dependent pathway?
- B). Is the induction of EMT reverted upon inhibition of the clathrin pathway?
- C). Is the TGF-beta stimulation of ECAD internalization clathrin dependent?

In response, we have performed the following experiments:

- A). We attempted to determine the pathway of EPN3-induced internalization of ECAD. To this extent, we performed KD of clathrin, of AP2 and treatment with dynasore (to inhibit dynamin). Both the KD of clathrin and AP2 required a minimum of 5 days after the first RNAi cycle in order to observe sufficient ablation of the proteins. Under these conditions MCF10A cells appeared suffering, in particular the clathrin KD cells (see attached Fig. 3). In the case of dynasore, the possibility of performing acute inhibition circumvented this problem.

Figure 3 for reviewers. Effects of clathrin KD or AP2 KD in MCF10A-Ctr and -EPN3. (A) MCF10A-Ctr or -EPN3 cells were silenced or not for clathrin or AP2 μ . IB was with the indicated antibodies. GAPDH was used as a loading control. MW markers are shown on the left. (B) Phase contrast microscopy of cells as in (A). Bar, 60 μ m.

We went ahead and measured ECAD internalization by IF. This methodology was used since dynasore-treated cells were rather sensitive to the steps required for the FACS-based endocytic assay and were lost during washes. By IF, conversely we were able to perform all experiments under the same conditions. The results of this set of experiments is attached here as Fig. 4 and can be summarized as follows:

- Dynasore treatment substantially affected ECAD endocytosis both in basal condition and upon EPN3 overexpression (see attached Fig. 4A). Thus, the pathway is dynamin-dependent.
- AP2 KD had no effect on the internalization of ECAD both in MCF10A control and in EPN3-expressing cells, while it almost completely inhibited endocytosis of Alexa488-transferrin in both cell systems (see attached Fig. 4A). These data argue that ECAD endocytosis in MCF10A is AP2-independent.
- Clathrin KD paradoxically caused an increase in the internalization of ECAD, while exerting the predicted inhibition of Tf internalization (see attached Figure 4A).

Figure 4 for reviewers. ECAD internalization is dynamin-dependent and AP2-independent. (A) Left, MCF10A-Ctr or EPN3 cells were silenced or not for clathrin or AP2 μ , or treated with dynasore 40 μ M for 30 min. Cells were subjected to EGF-starvation for 3 hrs and ECAD internalization was monitored in vivo by IF using an anti-ECAD antibody directed against its extracellular domain (HECD-1, Abcam). Antibody was allowed to internalize at 37 $^{\circ}$ C for 30 min. Cells were acid wash treated, fixed and stained with a secondary antibody to detect internalized ECAD (green). Blue, DAPI. Bar, 20 μ m. Right, box plot of relative ECAD fluorescence intensity for the indicated samples. P-value was calculated with Student's t-test. ***, <0.001;

***, <0.01; *, <0.05. **(B)** Right, MCF10A cells treated as in **(A)** were stimulated with Alexa488-transferrin for 30 min at 37°C, then were acid wash washed, to remove the non-internalized transferrin, and fixed. Blue, DAPI. Bar, 20 μm.

In addition, in the case of AP2 KD and clathrin KD, results were confirmed with the FACS-based assay, showing that clathrin KD increases ECAD endocytosis (attached Fig. 5), while AP2 KD had no effect (attached Fig. 5B). As mentioned, technical reasons precluded the performance of the FACS assay in dynasore-treated cells.

Figure 5 for reviewers. ECAD internalization by FACS upon KD of clathrin or AP2. **(A)** Time-course of ECAD internalization by FACS analysis in MCF10A-Ctr and -EPN3 cells, silenced or not for clathrin. Data are reported as fraction of mean fluorescence intensity of internalized ECAD signal over the total. **(B)** Time-course of ECAD internalization by FACS analysis in MCF10A-EPN3 cells silenced or not for clathrin or AP2μ. Data are reported as fraction of mean fluorescence intensity of internalized ECAD signal over the total, +/- SD of two independent experiments. The same experiment showed in **(A)** was averaged here for the samples MCF10A-EPN3 and MCF10A-EPN3 clat KD. P-value was calculated with Student's t test, two-tailed. *, <0.05.

We were rather puzzled by the results obtained with the clathrin KD also in light of the fact that morphologically the ECAD-containing vesicles, internalized in clathrin-KD cells, were quite different (smaller/more elongated) from those internalized in control cells (see attached Figure 4).

Thus, we performed additional investigations by treating clathrin-KD cells with dynasore (which inhibits ECAD internalization in control cells). Surprisingly, the internalization of ECAD, under conditions of clathrin KD, could not be inhibited by dynasore (see attached Figure 6). The most straightforward interpretation of these data is that the clathrin KD aberrantly upregulates a dynamin-independent pathway of ECAD that is not operative normally under unperturbed conditions.

Figure 6 for reviewers. Clathrin KD upregulates ECAD endocytosis through a dynamin-independent mechanism. (A) Left, MCF10A-Ctr cells were silenced or not for clathrin, and then treated or not with dynasore (40 μ M for 30 min). ECAD internalization was monitored in vivo by IF using an anti-ECAD antibody directed against its extracellular domain (HECD-1, Abcam). Antibody was allowed to internalize at 37°C for 30 min. Cells were acid wash treated, fixed and stained with a secondary antibody to detect ECAD (green). Blue, DAPI. Bar, 20 μ m. Right, box plot of relative ECAD fluorescence intensity for the indicated conditions. P-value was calculated with Student's t-test. ***, <0.001; **, <0.01.

As the reviewer will certainly realize, the sum of these results leaves us in the position of not being able to express a final judgment on the involvement of clathrin. What we can say is that the pathway is dynamin-dependent, AP2-independent. This is compatible with several forms of non-clathrin endocytosis, but also with clathrin-dependent/AP2-independent endocytosis (see for instance Pascolutti R *et al.*, Cell Report 2019).

We feel that trying to address the impact of clathrin would be technically rather challenging at this point. Our feeling is that this characterization, which we are interested in pursuing, would be better left to follow-up studies.

What we have done, in the revised version, is to clearly state that we have not determined yet the route of EPN3-stimulated internalization of ECAD (page 9 of the Results and pages 19-20 of the revised Discussion). We also discussed the fact that EPN3 physically interacts (albeit we do not know whether this is direct or indirect) with clathrin, but that this finding cannot be equaled to assume that it induces ECAD internalization via clathrin-mediated endocytosis, since EPN3 is also known to act redundantly with EPN1/2 in the clathrin-mediated internalization of other receptors (see also reply to previous point 3).

B). On the question of EMT and the clathrin pathway.

We have analyzed EMT markers under conditions of AP2-KD or clathrin-KD. The results are shown in the attached Figure 3.

By IB analysis, neither KD, could revert the EMT phenotype of EPN3 cells, and, at least in the case of AP2, there is a mild upregulation of NCAD (see attached Fig. 3A). This result seems to suggest that EPN3-induced EMT is clathrin- and AP2-independent.

Also in this case, However, in our opinion, the interpretation of this experiment remains problematic, mostly because data obtained with the clathrin KD would be subjected to the major interpretative caveat highlighted above.

C). On the question of TGF-beta stimulation of ECAD internalization

We performed ECAD internalization experiments in the presence of TGF-beta confirming that, also in this condition, ECAD internalization is dynasore-sensitive and AP2-independent, both in MCF10A-Ctr and -EPN3 expressing cells (attached Fig. 7). We did not perform experiments with the clathrin KD for the reasons mentioned above.

Also in this case, we cannot reach a final conclusion as to whether the ECAD endocytosis is due to clathrin-independent mechanism or to AP2-independent CME.

Figure 7 for reviewers. TGFβ-induced ECAD internalization is dynamin-dependent and AP2-independent. Left, MCF10A cells were silenced or not for AP2μ or treated with dynasore (40μM for 30 min). ECAD internalization was monitored in vivo by IF using an anti-ECAD antibody directed against its extracellular domain (HECD-1, Abcam). Antibody was allowed to internalize at 37°C for 30 min with or without of 5 ng/ml TGFβ. Cells were then acid wash treated, fixed and stained with a secondary antibody to detect ECAD (green). Blue, DAPI. Bar, 20 μm. Right, box plot of relative ECAD fluorescence intensity for the indicated conditions. P-value was calculated with Student's t-test. ***, <0.001; **, <0.01; *, <0.05.

In conclusion, the induction of a compensatory pathway of ECAD internalization, in the presence of clathrin-KD, did not enable us to reach a final conclusion on any of the three sub-points of the reviewer. What we would need here is the possibility of performing authentic acute inhibition of clathrin. We are presently attempting this, in collaboration with another group. These will not be simple experiments and we feel that they rather belong to follow-up studies.

REVIEWERS' COMMENTS:

Reviewer #1 (Remarks to the Author):

The authors very well addressed all of my concerns. I have no additional comments. They do not have to include the data using A549 cells.

Reviewer #2 (Remarks to the Author):

I thank you for your thorough responses to my concerns. The new information that you have added, has provided much more depth to the manuscript and its reported findings are now very much substantive and innovative in nature.

I have no additional comments or concerns.

Reviewer #3 (Remarks to the Author):

Revised version of manuscript NCOMMS-19-14370-T. The authors have very carefully addressed the points that I had raised on the initial version of the manuscript. For reasons that are diligently discussed in the rebuttal letter, the clathrin experiments have yielded results that cannot be fully understood at this stage. The work that's required for this will constitute a study on its own, and the authors have decided to explicit this aspect in the revised version of their current manuscript. I fully agree with the conclusions and interpretations that are presented by the authors, and truly think that the submission has been strengthened further by the revision process. The findings that are described in this manuscript very clearly are of major interest to a general life science readership.

Point-by-point reply to the Reviewers' comments:

Reviewer #1:

The authors very well addressed all of my concerns. I have no additional comments. They do not have to include the data using A549 cells.

R: We thank the Reviewer for his/her positive comment. As per the reviewer's indication, we did not include data on A459 cells in the final version of the manuscript.

Reviewer #2:

I thank you for your thorough responses to my concerns. The new information that you have added, has provided much more depth to the manuscript and its reported findings are now very much substantive and innovative in nature. I have no additional comments or concerns.

R: We thank the reviewer for the positive comment.

Reviewer #3:

The authors have very carefully addressed the points that I had raised on the initial version of the manuscript. For reasons that are diligently discussed in the rebuttal letter, the clathrin experiments have yielded results that cannot be fully understood at this stage. The work that's required for this will constitute a study on its own, and the authors have decided to explicit this aspect in the revised version of their current manuscript. I fully agree with the conclusions and interpretations that are presented by the authors, and truly think that the submission has been strengthened further by the revision process. The findings that are described in this manuscript very clearly are of major interest to a general life science readership.

R: We thank the reviewer for his/her positive revision.